# Hosts manipulate lifestyle switch and pathogenicity heterogeneity of opportunistic pathogens in the single-cell resolution

Ziguang Wang[1,2,3†], Shuai Li[4†], Sheng Zhang[1], Tianyu Zhang[5], Yujie Wu[1], Anqi Liu[1], Kui Wang[1], Xiaowen Ji[1], Haiqun Cao[1], Yinglao Zhang[1], Eng King Tan[6], Yongcheng Wang[5]*, Yirong Wang[4]*, Wei Liu[1]*

[1]School of Plant Protection; Anhui Province Key Laboratory of Crop Integrated Pest Management; Anhui Province Key Laboratory of Resource Insect Biology and Innovative Utilization, Anhui Agricultural University, Hefei, China; [2]College of Life Sciences, Nankai University, Tianjin, China; [3]First Clinical Medical College, Mudanjiang Medical College, Mudanjiang, China; [4]Bioinformatics Center, College of Biology, Hunan University, Changsha, China; [5]Liangzhu Laboratory, Zhejiang University, Hangzhou, China; [6]Department of Neurology, National Neuroscience Institute, Singapore General Hospital Campus, Singapore, Singapore

*For correspondence:
yongcheng@zju.edu.cn (YW);
wangyirong@hnu.edu.cn (YW);
liuwei5@ahau.edu.cn (WL)

†These authors contributed equally to this work

Competing interest: The authors declare that no competing interests exist.

**Abstract** Host-microbe interactions are virtually bidirectional, but how the host affects their microbiome is poorly understood. Here, we report that the host is a critical modulator to regulate the lifestyle switch and pathogenicity heterogeneity of the opportunistic pathogens *Serratia marcescens* utilizing the *Drosophila* and bacterium model system. First, we find that *Drosophila* larvae efficiently outcompete *S. marcescens* and typically drive a bacterial switch from pathogenicity to commensalism toward the fly. Furthermore, *Drosophila* larvae reshape the transcriptomic and metabolic profiles of *S. marcescens* characterized by a lifestyle switch. More importantly, the host alters pathogenicity and heterogeneity of *S. marcescens* in the single-cell resolution. Finally, we find that larvae-derived AMPs are required to recapitulate the response of *S. marcescens* to larvae. Altogether, our findings provide an insight into the pivotal roles of the host in harnessing the life history and heterogeneity of symbiotic bacterial cells, advancing knowledge of the reciprocal relationships between the host and pathogen.

## eLife assessment

The **fundamental** findings of this work substantially advance our understanding of the impact of the host on its gut microbes. The authors provided **compelling** evidence at single-cell resolution that the host can drive heterogeneity in the populations of gut microbes with significant consequences for the host physiology.

## Introduction

All metazoans ranging from insects to humans harbor a plethora of microbes referred to as the microbiome. The microbiome plays a pivotal role in host physiology and pathophysiology, with some species conferring benefits to the host and others causing damage (*Delannoy-Bruno et al., 2021*; *Lynch and Hsiao, 2019*; *Morais et al., 2021*). Over the past decades, tremendous effort, including

ours (*Jia et al., 2021*; *Liu et al., 2022*; *Liu et al., 2017*), has been devoted to understanding the impact of microbial strains or more complex communities on their hosts. In fact, interactions between the host and microbiome are mutually bidirectional (*Bäckhed et al., 2005*; *Jiang et al., 2019*), conferring benefits to the host and microbial sides. Nevertheless, the knowledge of the effect of the host on the resident microbial community is still in its infancy. Studies reveal that hosts play a crucial role in shaping the assembly and composition of their unique microbiome (*Muegge et al., 2011*; *Olm et al., 2022*; *Valeri and Endres, 2021*). However, environmental fluctuations frequently happen at much shorter time scales that preclude bacterial adaptation by genetic mutation or species displacement. To cope with these situations, microbial communities have developed sophisticated transcriptional reprogramming to globally regulate gene expression (*Avraham et al., 2015*; *Becattini et al., 2021*; *Prescott and Decho, 2020*). Pathobionts routinely sense and specialize on host-derived substrates, execute context-dependent transitions from harmful to commensal states, and generate the host-associated continuum (*Barak-Gavish et al., 2023*; *Proctor et al., 2023*; *Somvanshi et al., 2012*). In this regard, the molecular mechanism by which the host restrictively controls gene transcription and metabolism of their microbiome remains much undefined.

Microbial populations frequently aggregate in several dozen micrometers and stochastically undergo specific differentiation into subpopulations with strikingly distinct properties, employing a strategy known as bet-hedging to survive rapid changes in the environment (*Eldar and Elowitz, 2010*; *Raj and van Oudenaarden, 2008*). As such, bacteria manifest heterogeneous transcriptional profiles, leading to phenotypic heterogeneity among individual bacterial cells (*Dar et al., 2021*; *Gasch et al., 2017*). Traditionally, gene expression of bacterial cells has been investigated in bulk or on a population level by mixing and averaging mRNA simultaneously from sorts of cells. Single-cell transcriptomics is revolutionizing the analysis of phenotypic cell-to-cell variation in eukaryotes, enabling us to explore the heterogeneity heretofore hidden within population behavior (*Klein et al., 2015*; *Mancio-Silva et al., 2022*; *Perez et al., 2022*). Recently, droplet-based high-throughput bacterial single-cell RNA sequencing (RNA-seq) methods have been developed and applied to study antibiotic-associated heterogeneous cellular states (*Ma et al., 2023*; *Xu et al., 2023*). We took advantage of this approach to tackle the challenge that the host induces transcriptionally distinct subpopulations of target bacteria during the interaction, contributing to a full appreciation of the phenotypic heterogeneity within microbiome.

*Drosophila* has a relatively simple bacterial community of commensal and opportunistic pathogens, typically predominated by 5–20 bacterial species. All *Drosophila* microbes are facultative bacteria that alternate between free-living and host-associated lifestyles. *Serratia marcescens* is an opportunistic pathogen and generates a pink pigment prodigiosin that is characteristically oscillatory to metabolic activities, making it a visible bioindicator of *S. marcescens* metabolism. In this regard, *Drosophila* has provided a promising model to study symbiosis, dysbiosis (*Chandler et al., 2011*; *Kim et al., 2020*).

To investigate whether and/or how the host modulates bacterial lifestyles, we used a reductionist approach in which germ-free (GF) *Drosophila* was re-associated with *S. marcescens* (*Kuraishi et al., 2011*). We found that *Drosophila* larvae sufficiently outcompeted *S. marcescens*, and resulted in shifts in the transcriptomic and metabolomic profile in bulk and single-cell resolution, providing a robust paradigm to further study the host-microbe interaction with the *Drosophila* model.

## Results

### *Drosophila* larvae alter surface topography and population size of microbes

All *Drosophila* microbes are facultative bacteria that alternate between free-living and host-associated lifestyles. Free-living bacteria initially form small microcolonies extending from the substratum, continue to expand, coalesce, and eventually generate a surface slick (a biofilm-like cell mat) that is visible to the naked eye (*Koo and Yamada, 2016*). Fascinatingly, the topography of surface slick was typically altered by fly colonization. The surface slick associated with strong wild-type flies was corn-like yellow, and the mature layer of the bacterial community was completely broken (*Figure 1A*). The destruction process was further exacerbated by overcrowded larvae that liquefied the upper food layer, leading to a yellow aqueous layer (*Figure 1B*). However, it was gray and partially segmented by weak flies (*Figure 1C*), and even pale and intact associated with infertile flies (*Figure 1D*). These

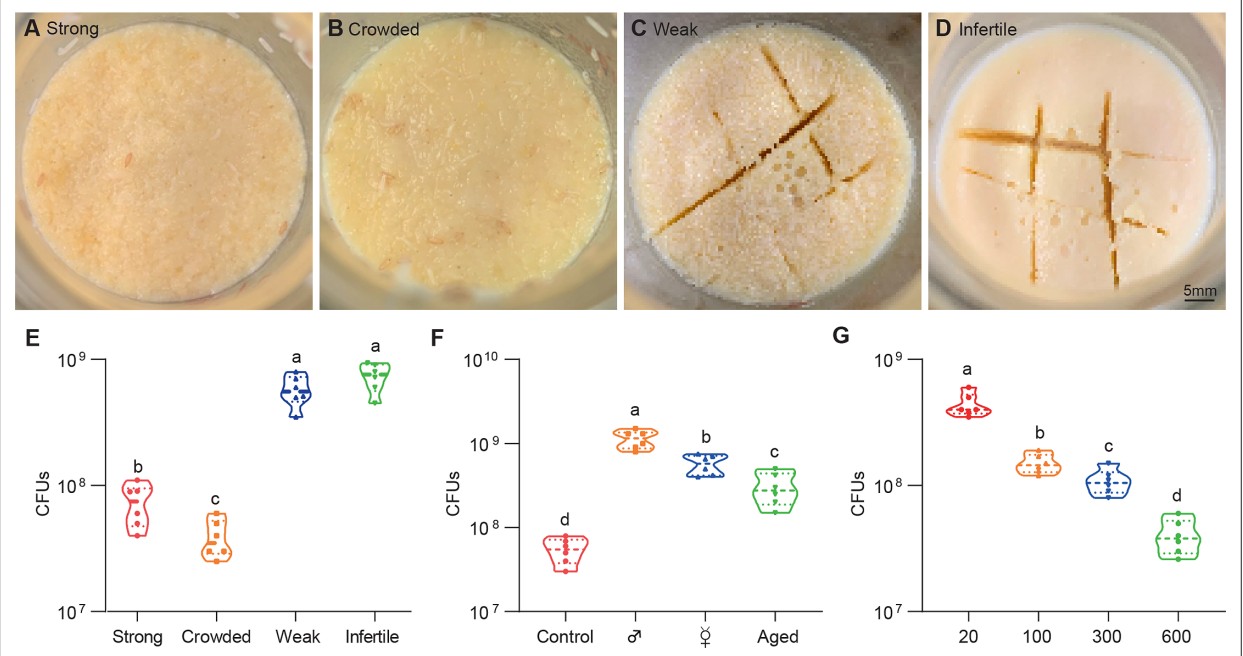

**Figure 1.** *Drosophila* larvae shape topographies and carrying capacities of bacterial community. (**A–D**) Representative images of sticky 'biofilm-like' formation on the surface of the sugar-corn-yeast medium whereby *Drosophila* flies with differential robustness were raised. The topographies of surface slick are differentially deconstructed and segmented by flies with different robust flies. (**E**) Bacterial loads of the diet associated with strong, crowed, weak, and infertile flies, respectively. Strong flies (wild-type fly CS), weak flies (*yw; Sp/CyO; MKRS/TM6B*), infertile flies (*dfmr1⁵⁰ᴹ* null mutant). n=6 for each. (**F**) Bacterial loads of the diet associated with control, male, virgin, and aged flies, respectively. Each bottle contained 100 flies with a 1:1 ratio of male:female in control and aged groups, and 50-day-aged flies were used. n=6 for each. (**G**) Bacterial loads of the diet associated with *Drosophila* larvae in a dosage-dependent manner. n=6 for each. Means ± SEMs. All variables have different letters, and they are significantly different (p<0.05). If two variables share a letter, they are not significantly different (p>0.05). Kruskal-Wallis test followed by Dunn's multiple comparisons test.

results indicate that *Drosophila* plays a critical role in altering the topography of bacterial community on *Drosophila* media (***Deines et al., 2020***). Our data showed that the cultivable bacterial loads associated with weak or infertile flies were dramatically higher than those of strong stocks (***Figure 1E***), indicating that the host diminishes the bacterial load of the shared niche.

Given that *Drosophila* larvae and adults coexist in the bottles, we sought to determine which were directly responsible for these alterations. Similarly, bacterial loads were substantially higher in the medium associated with males and virgin females (without progenies) than in the control (***Figure 1F***), suggesting that larvae mainly cause a decline in bacterial population size. Consistently, the 50-day-aged flies that deposited few eggs mimicked weak flies with gray surface slick and a decline in bacterial loads. Moreover, transplanted larvae directly caused a dose-dependent decline in the overall bacterial loads in the habitat (***Figure 1G***). These findings demonstrate that *Drosophila* larvae play a critical role in outcompeting their symbionts in the shared habitat.

### *Drosophila* larvae antagonize *S. marcescens* in the niche

To experimentally characterize the impact of the host on the symbiont, we applied a reductionist approach in which *Drosophila* mono-associated with *S. marcescens* was generated as depicted in ***Figure 2A***. Crawling larvae (~96 hr post oviposition) were concurrently transferred to fly food vials with $10^7$ CFU bacterial load. We found that *S. marcescens* alone formed a pink surface slick in the medium over time (***Figure 2B***, top). The color intensity of surface slicks was initially accumulated, peaked at 24 hr timepoint post inoculation, but gradually faded thereafter. We quantified the optical density of prodigiosin inside surface slicks with the spectrophotometer as described (***Kalivoda et al., 2010***; ***Pan et al., 2020***). The time-course amount of prodigiosin was congruent with the color density visible to the naked eye above (***Figure 2C***). Compared to *S. marcescens* alone, the color intensity of surface slicks was significantly dampened by larval transplantations (***Figure 2B***, bottom). As expected, the amount of prodigiosin of *S. marcescens* in coculture substantially declined compared with that of

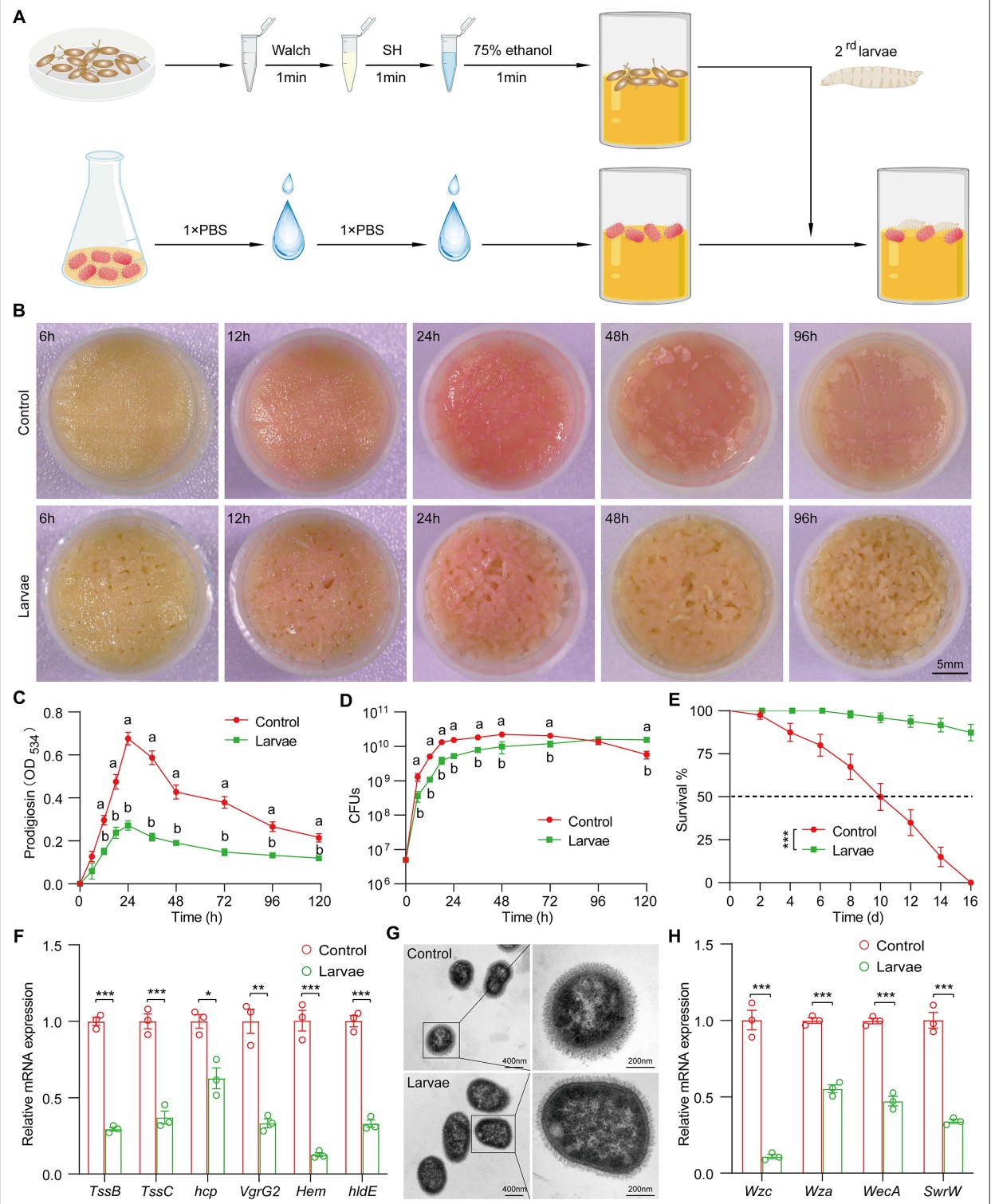

**Figure 2.** *Drosophila* larvae outcompete *S. marcescens* in the diet. (**A**) A diagram of a reductionist approach to investigate the role of *Drosophila* in regulating the physiology and behavior of *S. marcescens*. Top: Germ-free (GF) *Drosophila* larvae were generated by successive sterilization of fresh eggs with sanitizer Walch, sodium hypochloride (SH), ethanol, and PBS containing 0.01% Triton X-100 (PBST). Bottom: *S. marcescens* was cultured in a liquid medium and re-inoculated to fly cornmeal food after washing with PBS buffer. In the meantime, GF crawling larvae were transferred to the fly medium in the shared vials with *S. marcescens*. (**B**) Representative images of surface slick inoculated with *S. marcescens* alone and with *S. marcescens* over time. (**C**) The prodigiosin production of *S. marcescens* alone and in coculture at different timepoints. Prodigiosin production was assessed with the spectrometer at $OD_{534}$. n=6 for each. (**D**) Bacterial loads of *S. marcescens* alone and in coculture in the time course. n=6 for each. (**E**) The survival

*Figure 2 continued on next page*

*Figure 2 continued*

rate of adult flies challenged with *S. marcescens* alone and in coculture. The bacterial suspension (20 μl, OD$_{595}$=1) was supplemented to a vial with autoclaved fly food. Single and coculturing *S. marcescens* were obtained after 24 hr incubation as described in A, and the percentage of living female flies treated with *S. marcescens* alone and in coculture was calculated to monitor lifespan. n=180 for each. The statistical analyses were performed using log-rank test. ***p<0.001. (**F**) RT-qPCR analysis of the expression levels of virulence-related genes of *S. marcescens* alone and in coculture. n=3 for each. (**G**) Transmission electron microscopy of *S. marcescens* alone and in coculture. Scale bars: 400 nm (left panel) or 200 nm (right panel). (**H**) RT-qPCR analysis of the expression levels of extracellular polysaccharide production-related genes in the control and larvae groups. n=3 for each. Means ± SEMs. The statistical analyses were performed using two-tailed unpaired Student's t-test in (**F**) and (**H**). *p<0.05; **p<0.01; ***p<0.001. All variables have different letters, and they are significantly different (p<0.05). If two variables share a letter, they are not significantly different (p>0.05). Kruskal-Wallis test followed by Dunn's multiple comparisons test.

The online version of this article includes the following figure supplement(s) for figure 2:

**Figure supplement 1.** *Drosophila* larvae modulate *S. marcescens* lifestyle switch.

**Figure supplement 2.** Biological factors mainly determine *S. marcescens* lifestyle.

*S. marcescens* alone. Next, we examined bacterial titers in the food at regular intervals. Our result showed that single *S. marcescens* grew more rapidly than coexisting *S. marcescens*, and the population of single *S. marcescens* reached the plateau value within 1 day (*Figure 2D*), indicating that the niche possesses a finite carrying capacity of bacteria. However, the population of coexisting *S. marcescens* reached the plateau value within 2 days. Moreover, the number of CFUs was predominantly suppressed by larva transplantation at the initial phases of colonization (within 72 hr post inoculation) compared to that of *S. marcescens* alone (*Figure 2D*), consistent with the result that larvae thwarted the number of total mixed bacteria in the diet (*Figure 2B*). Noteworthily, the growth of single *S. marcescens* was comparable (at 96 hr post inoculation) or even lower (at 120 hr post inoculation) than corresponding *S. marcescens* in coculture, likely that single *S. marcescens* rapidly exhausted their nutritional resources and underwent ecological suicide (*Ratzke et al., 2018*). These results suggest that the larva acts as a potential competitor that efficiently prevents the overgrowth of *S. marcescens* in the habitat before 72 hr post inoculation. To verify it, different numbers of larvae were added to vials inoculated with *S. marcescens*. Indeed, the more were larvae, the less the color intensity of the surface slick (*Figure 2—figure supplement 1A and B*). In addition, the population size of *S. marcescens* in food was in restricted control by larvae in a dosage-dependent manner (*Figure 2—figure supplement 1C*). Altogether, the cumulating results suggested that *Drosophila* larva has a competitive advantage in the habitat, and acts as a critical regulator of *S. marcescens*.

### *Drosophila* larvae modulate the pathogen-to-commensal transition of *S. marcescens*

*S. marcescens* is a *Drosophila* pathobiont with the potential to switch between commensalism and pathogenicity toward the host. To this end, we sought to examine the *S. marcescens* lifestyle switch from pathogenicity to commensalism by assessing the respective survival of flies on the fly medium that had been processed by single or coexisting *S. marcescens*. Our data showed that flies challenged with *S. marcescens* alone manifested higher mortality than flies with *S. marcescens* in coculture (*Figure 2E*), suggesting that larva antagonizes the pathogenicity of *S. marcescens*. In addition, the expression of virulent factors was thwarted by larvae (*Figure 2F*). However, *S. marcescens* more efficiently sustained optimal larval development upon nutrient scarcity than both axenicity and commensal *L plantarum* (*Matos et al., 2017*; *Storelli et al., 2011*; *Figure 2—figure supplement 1D*), indicating that larvae-associated *S. marcescens* could impart fitness benefits to their hosts. To rule out the possibility that phenotypic alterations could stem from genomic mutations, we examined the prodigiosin yield and CFUs of re-culturing *S. marcescens* that had coexisted with larvae. Our results showed that neither prodigiosin yield nor CFUs of re-culturing *S. marcescens* differed from the original strain (*Figure 3—figure supplement 1A–C*), suggesting that a phenotypic switch was driven primarily by transcriptional reprogramming. The cell wall of Gram-negative bacteria is composed of a single layer of peptidoglycan surrounded by a membranous structure, which may function as a virulence factor (*Chu et al., 2016*; *Pan et al., 2022*). Indeed, transmission electron microscopy (TEM) confirmed a significant reduction in the width of the cell wall of *S. marcescens* in coculture compared to that of *S. marcescens* alone (*Figure 2G*). Additionally, we observed that the expression of genes related to cell wall biosynthesis was dampened by larva transplantation (*Figure 2H*). Taken together, these results

suggest that *Drosophila* larvae modulate *S. marcescens* lifestyle from pathogenicity to commensalism toward the host.

Presumably, surface slicks can be destroyed by mechanical force from larva crawling and burrowing (***Dufrêne and Persat, 2020***). To tackle this issue, we agitated fly food to imitate larva locomotion (***Figure 2—figure supplement 2A***). Indeed, agitation decreased color intensity, prodigiosin yield, and population size in fly food (***Figure 2—figure supplement 2B–D***), implying that mechanical force accounts for the negative regulation of *S. marcescens*. However, larval transplantation resulted in a much more robust decline in color intensity of surface slicks, prodigiosin yield, and population size than agitation alone. Of note, the surface of the slick with agitation appeared lighter than that of larvae, possibly due to a stratification of prodigiosin following agitation. In addition, the expression of prodigiosin synthesis genes was thwarted by the agitation (***Figure 2—figure supplement 2E***). These results indicated that larva-derived biofactors and/or synergism of force and biofactors primarily confer the inhibition of *S. marcescens* (see the later results).

## *Drosophila* enforces bacterial global transcriptional and metabolic adaptation to the host

In order to understand the molecular basis for the bacterial lifestyle switch from pathogenicity to commensalism in response to *Drosophila* larvae, bulk RNA-seq analysis was applied to bacterial cells 24 hr after inoculation. We devised an approach to efficiently collect bacteria from the agar fly food as described in Materials and methods. As shown in ***Figure 3A***, principal component analysis (PCA) showed a larvae-dependent separation to *S. marcescens* alone on the first principal component of variance. Compared to single *S. marcescens*, larvae-associated *S. marcescens* exhibited significant upregulation of 360 genes and downregulation of 439 genes (***Figure 3B*** and ***Supplementary file 2*** Table S2), respectively. Functional annotations of the differentially expressed genes (DEGs) were assigned using the Kyoto Encyclopedia of Genes and Genomes (KEGG). KEGG pathway enrichment analysis highlighted that the most differentially upregulated DEGs of coexisting *S. marcescens* were related to bacterial proliferation and growth, including ribosome, translation factors, transcription, DNA replication proteins, and energy metabolism (***Figure 3D***), implying that larval transplantations favored maintenance of *S. marcescens* in the diet. By contrast, the most differentially downregulated DEGs of single *S. marcescens* were related to bacterial pathogenicity, including transporters, excretion, quorum sensing, and exosome (***Figure 3C***). To validate our findings in bulk RNA-seq analysis, quantitative PCR (qPCR) was used to examine the expression of predicted genes involved in bacterial proliferation and pathogenicity. Indeed, most predicted genes associated with pathogenicity were downregulated in single *S. marcescens* (***Figure 3E***), while genes associated with bacterial proliferation were upregulated in response to larva colonization (***Figure 3F***). Consistent with the previous result that this phenotypic switch was driven by transcriptional changes, the expression of virulent and growth genes was recovered after re-culturing (***Figure 3—figure supplement 1D and E***). Overall, *Drosophila* forms a symbiosis with the bacterial community by harnessing bacterial global transcription.

Next, we investigated whether larvae could further elicit changes in the metabolism of *S. marcescens* using untargeted metabolomics. The data showed that 91 metabolites were identified and their concentrations were quantified (Table S3 ***Supplementary file 3***. Metabolomic profiles discriminated larva-associated *S. marcescens* from *S. marcescens* alone (***Figure 4A***), indicating that the host reshapes the global metabolic profile of bacterial cells. We found that larva-associated *S. marcescens* displayed significant upregulation of 22 metabolites and downregulation of 69 metabolites (***Figure 4B***). Unsupervised hierarchical clustering of the metabolome revealed distinct clusters of metabolites in *S. marcescens* alone versus coculture (***Figure 4C***). Moreover, we detected a significant decrease in amino acid metabolism, phosphotransferase system, and ABC transporters in *S. marcescens* in coculture compared to *S. marcescens* alone (***Figure 4D***). Consistent with the previous studies (***Defoirdt, 2019***; ***Wen et al., 2022***), these results suggested that the host suppresses differentiation of *S. marcescens* into the population with pathogenicity. By contrast, the most differentially upregulated metabolites of *S. marcescens* in coculture were related to the biosynthesis of fatty acids and unsaturated fatty acids, and the pentose phosphate pathway (***Figure 4E***). Taken together, these results suggest that the host affects the global metabolic profile of symbiotic cells and drives the switch of the bacterial lifestyle from pathogenicity to commensalism.

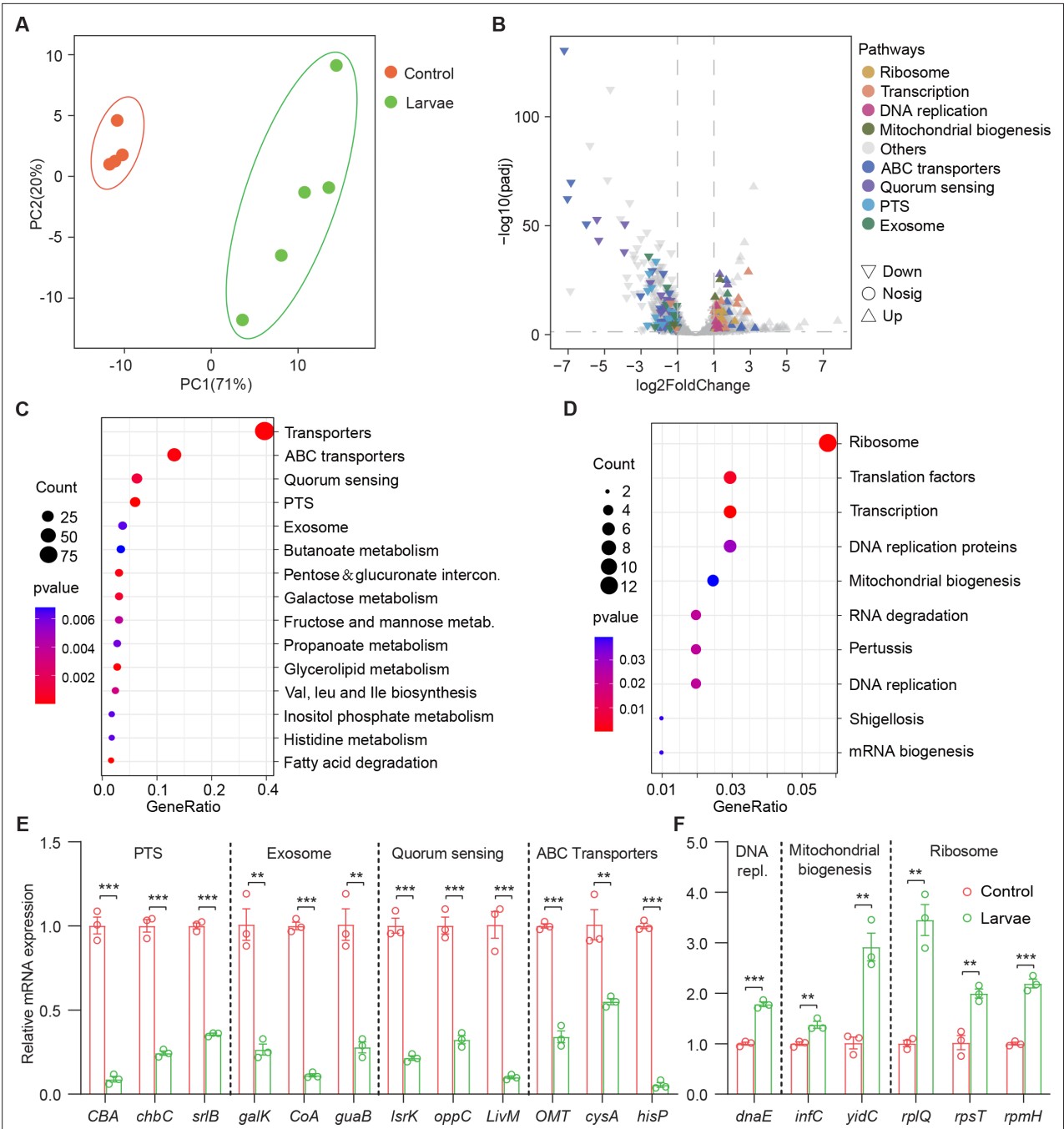

**Figure 3.** *Drosophila* larvae adjust bacterial global transcriptional adaptation to the host. (**A**) Principal component analysis (PCA) of unweighted, jack-knifed UniFrac distances of the transcriptional profile of *S. marcescens* alone and with larvae. PC1, principal component 1; PC2, principal component 2. Scattered dots in different colors represent samples from different experimental groups. n=4–5. (**B**) Volcano plot comparing gene expression profiles of *S. marcescens* alone and with larvae after 24 hr of incubation. X-axis represents the log2-transformed value of gene expression change folds between larvae and control groups. Y-axis represents the logarithmic transformation value of gene expression levels in *S. marcescens*. Genes belonging to different pathways are represented by different colored shapes as indicated. ▽ depicts genes significantly upregulated in *S. marcescens* with larvae compared to *S. marcescens* alone (log2 fold change<1; adjusted p<0.01), and △ depicts genes significantly downregulated in *S. marcescens* with larvae (log2 fold change<1; adjusted p<0.01) compared to *S. marcescens* alone. O depicts genes without significant alteration compared to *S. marcescens* alone. (**C, D**) Kyoto Encyclopedia of Genes and Genomes (KEGG) analysis of the significantly downregulated and upregulated genes in *S. marcescens* with larvae compared to *S. marcescens* alone. (**E, F**) RT-qPCR analysis of the expression levels of downregulated and upregulated genes in the control and larvae groups. n=3 for each. Means ± SEMs. The statistical analyses were performed using two-tailed unpaired Student's t-test. **p<0.01; ***p<0.001.

*Figure 3 continued on next page*

*Figure 3 continued*

The online version of this article includes the following figure supplement(s) for figure 3:

**Figure supplement 1.** The lifestyle switch of *S. marcescens* was driven by transcriptional alterations.

The co-expression network analysis of transcriptome and metabolome revealed that 42 differentially expressed metabolites were found to be related to the pathogen-to-commensal switch, including ribosome, transcription, DNA replication, energy metabolism, ABC transporters, phosphotransferase system, quorum sensing, and exosome (*Figure 4—figure supplement 1A*). Serotonin, sorbitol, and lactobionic acid were related to the pathogen-to-commensal switch (p-value<0.05). To validate these results, we perturbed the *S. marcescens* extracellular environment by adding predicted metabolites to the fly food. Indeed, we found that serotonin, sorbitol, and lactobionic acid efficiently reduced the prodigiosin yield and CFUs (*Figure 4—figure supplement 1B and C*), highlighting the high quality of the prediction. Moreover, *S. marcescens* perturbed with sorbitol manifested less virulent to flies (*Figure 4—figure supplement 1D*) and the impaired transcription of virulence-related genes (*Figure 4—figure supplement 1E*), resembling the lifestyle of co-culturing *S. marcescens*. Collectively, these findings suggest that *Drosophila* enforces bacterial transcriptional and metabolic adaptation to the host.

## Larvae modulate pathogenicity heterogeneity of *S. marcescens*

Isogenic microbial populations can generate transcriptional variation across individual cells, thereby prompting us to analyze bacterial populations at the single-cell resolution. We characterized the transcriptome of individual *S. marcescens* cells by implementing bacterial single-cell RNA-seq on a platform available. To validate this approach, we first determined the capacity of this platform to distinguish bacterial populations of heat-shocked *S. marcescens* grown in a liquid medium as previously described (*Dar et al., 2021*). Indeed, the results provide strong evidence that the bacterial single-cell RNA-seq approach used was sufficient and reliable to capture transcriptional responses to heat shock (*Figure 5—figure supplement 1*).

Next, we speculated about the heterogeneous transcriptional response of *S. marcescens* challenged with larvae as well as mechanical force. The growth of *S. marcescens* was well understood in fly food conditions as described in *Figure 2*, so we collected *S. marcescens* at the late exponential phase when certain bacterial cells manifested their differentiation and pathogenicity. We captured 2800 cells in total and detected a median of 198 genes per cell for control, 333 for force, and 175 for larvae (*Figure 5A*). Strikingly, bacterial cells from each group tended to form three distinct clusters that correspond to the agitated, larva-associated, and control cultures could be visualized by graph-based clustering of their gene expression profiles (*Figure 5B*). This result suggested that larval transplantation, as well as agitation, induce significant changes in gene expression patterns, consistent with the bulk RNA-seq result that larvae caused the shift in bacterial global transcription using Seurat (*Kuchina et al., 2021*). Next, we attempted to partition the total bacterial population of *S. marcescens* into subpopulations with diverse predicted functional capabilities by the algorithmic grouping of cell-expression profiles. As shown in *Figure 5C*, we further partitioned the three groups into seven subclusters (Subclusters 0–6) representing subpopulations with distinct expression profiles, while the gene expression in each cluster was continuous. Namely, the sampled populations of *S. marcescens* alone and in coculture were respectively partitioned into three coexisting subgroups (Subclusters 0, 1, and 2 for co-culturing *S. marcescens* and Subclusters 3, 4, and 6 for single *S. marcescens*), implicating that phenotypic diversity is a general feature of bacteria in clonal populations.

To understand the heterogeneity of gene expression patterns, we identified the DEGs associated with pathogenicity to characterize the transcriptional profile (*Supplementary file 4* Table S4), and ranked marker genes in each cluster (*Figure 5D*). We selected pathogenicity-related DEGs, including *livI* (ABC transporter), *oppA* (quorum sensing), *secY* (secretion system), and *fp* (virulence), and charted the feature expression of them in each cluster in low-dimensional space (*Figure 5E, F, I, J*). We found most genes involved in pathogenicity displayed a substantial decline upon larvae, which was consistent with the previous findings of reduced pathogenicity of coexisting *S. marcescens*. Moreover, we found that *livI* exhibited gradually higher expression down along Subcluster 3 to Subcluster 6 (*Figure 5G*), implicating the potentially hierarchical virulence-regulatory network in these three subclusters. Analogously, *oppA*, *secY*, and *fp* exhibited a similar expression pattern in these three

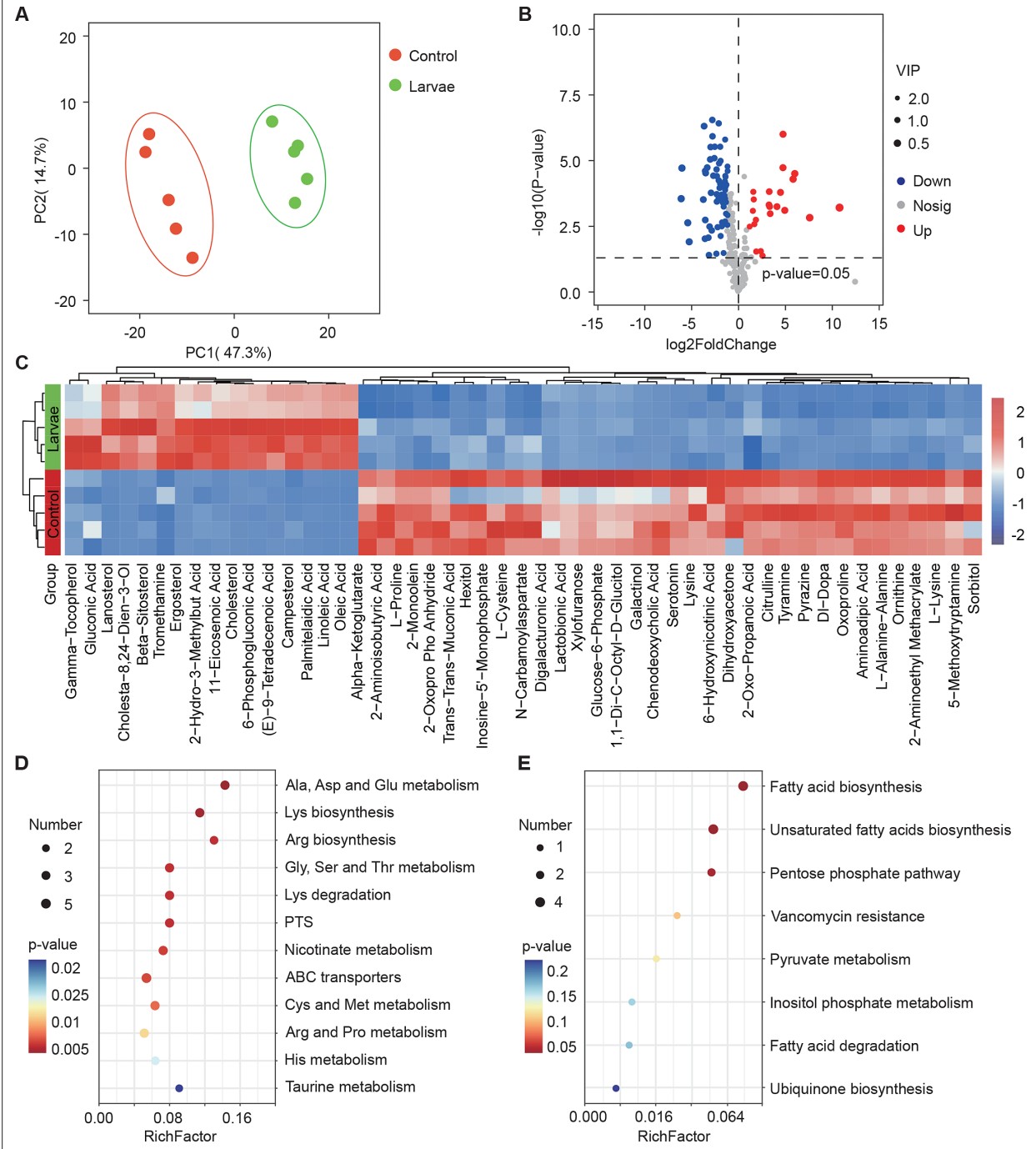

**Figure 4.** *Drosophila* larvae affect the global metabolic profile of *S. marcescens*. (**A**) Principal component analysis (PCA) of unweighted, jack-knifed UniFrac distances of metabolic profile of *S. marcescens* alone and with larvae. PC1, principal component 1; PC2, principal component 2. Scattered dots in different colors represent samples from different experimental groups. n=5 for each. (**B**) Volcano plot comparing metabolic profiles of between control and larvae groups after 24 hr of incubation. X-axis represents the log2-transformed value of gene expression change folds between larvae and control groups. Y-axis represents the logarithmic transformation value of gene expression levels in *S. marcescens*. Red dots depict genes significantly upregulated in *S. marcescens* with larvae compared to *S. marcescens* alone (log2 fold change<1; adjusted p<0.01), and blue dots depict genes significantly downregulated in *S. marcescens* with larvae (log2 fold change<1; adjusted p<0.01) compared to *S. marcescens* alone. Gray dots depict genes without significant alteration compared to *S. marcescens* alone. (**C**) The distinct clusters of metabolites in *S. marcescens* alone versus coculture. (**D, E**) Kyoto Encyclopedia of Genes and Genomes (KEGG) pathway analysis of the significantly downregulated and upregulated metabolites in *S. marcescens* with larvae compared to *S. marcescens* alone.

The online version of this article includes the following figure supplement(s) for figure 4:

**Figure supplement 1.** Interaction network analysis of transcriptome and metabolome.

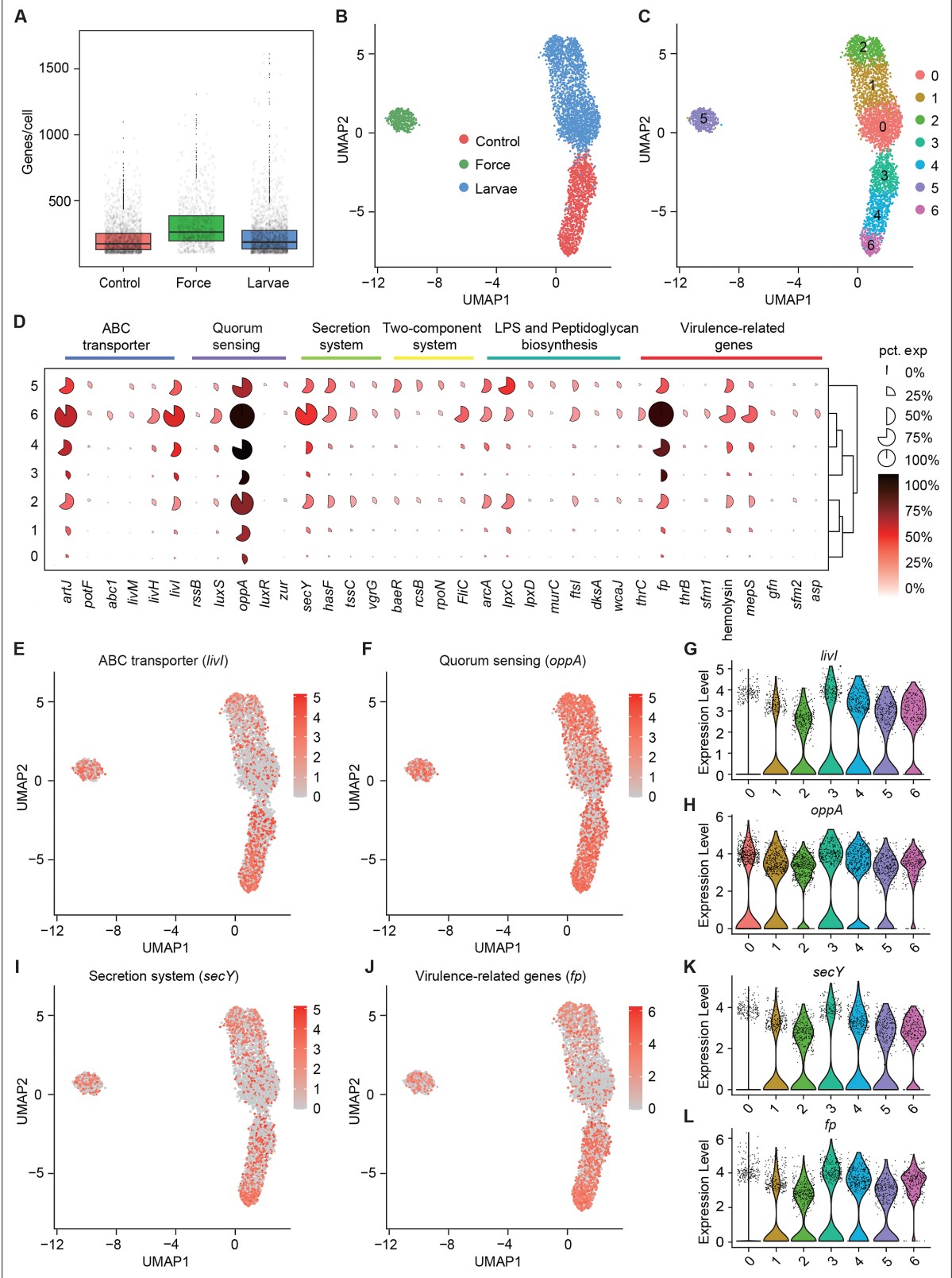

**Figure 5.** Pathogenicity heterogeneity of *S. marcescens*. (**A**) mRNA gene counts per cell for *S. marcescens* alone, with force and with larvae. Each dot represents a bacterial cell of *S. marcescens*. (**B**) Joint UMAP two-dimensional analysis showing that are distinct clusters among *S. marcescens* alone, with force and with larvae. (**C**) The cell subpopulation among the control, force, and larvae groups. There were three distinct subpopulations in the control and force groups. (**D**) Mean expression levels of genes involved in ABC transporter, quorum sensing, secretion system, two-component system,

*Figure 5 continued on next page*

*Figure 5 continued*

LPS and peptidoglycan biosynthesis, and virulence-related genes in different subclusters. The shape of each dot indicates the proportion of cells in the cluster, while the color indicates the average activity normalized from 0% to 100% across all clusters. (**E, F**) The expression of a representative gene of ABC transporter and quorum sensing was highlighted on the UMAP. The red color bars represent the normalized expression of a gene across all cells analyzed. (**G, H**) Violin plots of *livI* and *oppA* gene in different subclusters. Each dot represents a single cell and the shapes represent the expression distribution. (**I, J**) The expression of a representative gene of secretion system and virulence-related genes was highlighted on the UMAP. The red color bars represent the normalized expression of a gene across all cells analyzed. (**K, L**) Violin plots of *secY* and *fp* gene in different subclusters. Each dot represents a single cell and the shapes represent the expression distribution.

The online version of this article includes the following figure supplement(s) for figure 5:

**Figure supplement 1.** Heat-induced phenotypic heterogeneity of *S. marcescens*.

subclusters (*Figure 5H, K, and L*). Given of increased pathogenicity along three subclusters, it was likely that Subcluster 6 could be developed from Subcluster 4, thereby inferring that Subcluster 6 could be identified as subpopulations of single *S. marcescens* in a bona fide pathogen. Intriguingly, many virulence-associated genes were highly expressed in Subclusters 5, implying that mechanical force could not attenuate the virulence of *S. marcescens*. We tested it by assessing the survival of flies challenged with processed fly medium from agitated *S. marcescens*. Indeed, flies treated with agitated *S. marcescens* displayed a significantly shorter lifetime than flies with coculturing *S. marcescens* (*Figure 6—figure supplement 1A*). Taken together, our results demonstrated the presence of pathogenicity heterogeneity of *S. marcescens* at the single-microbe-level transcriptional landscape.

## Larvae modulate growth heterogeneity of *S. marcescens*

We showed that coexisting *S. marcescens* were related to growth, so we continued to identify the DEGs associated with bacterial propagation to characterize the transcriptional profile (*Supplementary file 4*Table S4), and ranked marker genes in each cluster (*Figure 6A*). Among DEGs of ribosome between single *S. marcescens* and co-culturing *S. marcescens*, *rpsL* encodes the highly conserved rps12 protein of the ribosomal accuracy center, while, tryptophan, biosynthesized by *trpD* among DEGs of DNA replication, is an essential nutrient and serves as a building block for protein synthesis and DNA replication. As anticipated, most genes involved in growth showed an evident increase in larvae. Interestingly, *rpsL* and *trpD* similarly exhibited gradually higher expression up along Subcluster 0 to Subcluster 1 (*Figure 6B–E*), indicating the potentially hierarchical growth-regulatory network in these three subclusters. Taken together, our results demonstrated the presence of bacterial propagation heterogeneity of *S. marcescens* at the single-microbe-level transcriptional landscape.

Nitrogen is the fundamental element of nucleic acids and proteins. In the clusters corresponding to larvae (Clusters 0, 1, and 2), we observe peak expression of genes involved in nitrogen metabolism, such as P-II family nitrogen regulator (*glnK*) and glutamate-ammonia ligase (glnA; *Figure 6F and H* and *Figure 6—figure supplement 1B and D*). *glnK* plays a critical role in regulating the activity of glutamine synthetase (e.g. *glnA*), which promotes glutamine synthesis. Glutamine is the most abundant non-essential amino acid, and serves as nitrogen and carbon sources for cell growth and differentiation. Moreover, these three subclusters showed a significant increase in the activity of genes involved in urea transport and utilization, such as urea ABC transporter substrate-binding protein (*urtA*), allophanate hydrolase (*atzF*), and urea carboxylase (*uca*). *urtA* accounts for the transport of urea, and then *atzF* and *uca* synergize to convert urea to ammonia that can be used for glutamine biosynthesis. Additionally, glutamine synthesis needs a large amount of energy, so we then turned to carbon metabolism. Indeed, we observed higher expression of phosphoglycerate kinase (*pgk*) and bifunctional acetaldehyde-CoA/alcohol dehydrogenase (*adhE*) that participate in glycosis in the Subclusters 0, 1, and 2 (*Figure 6G* and *Figure 6—figure supplement 1C*). Moreover, we found that *pgk* and *adhE* exhibited gradually higher expression in these three subclusters relative to subclusters with single *S. marcescens* (*Figure 6I* and *Figure 6—figure supplement 1E*), suggesting that bacteria could establish a dedicated replicative niche to efficiently replicate. The regulatory schematic is shown in *Figure 6J*. Altogether, these findings that the host globally results in transcriptional reprogramming of single-cell *S. marcescens*, facilitating the cooperation among individual cells.

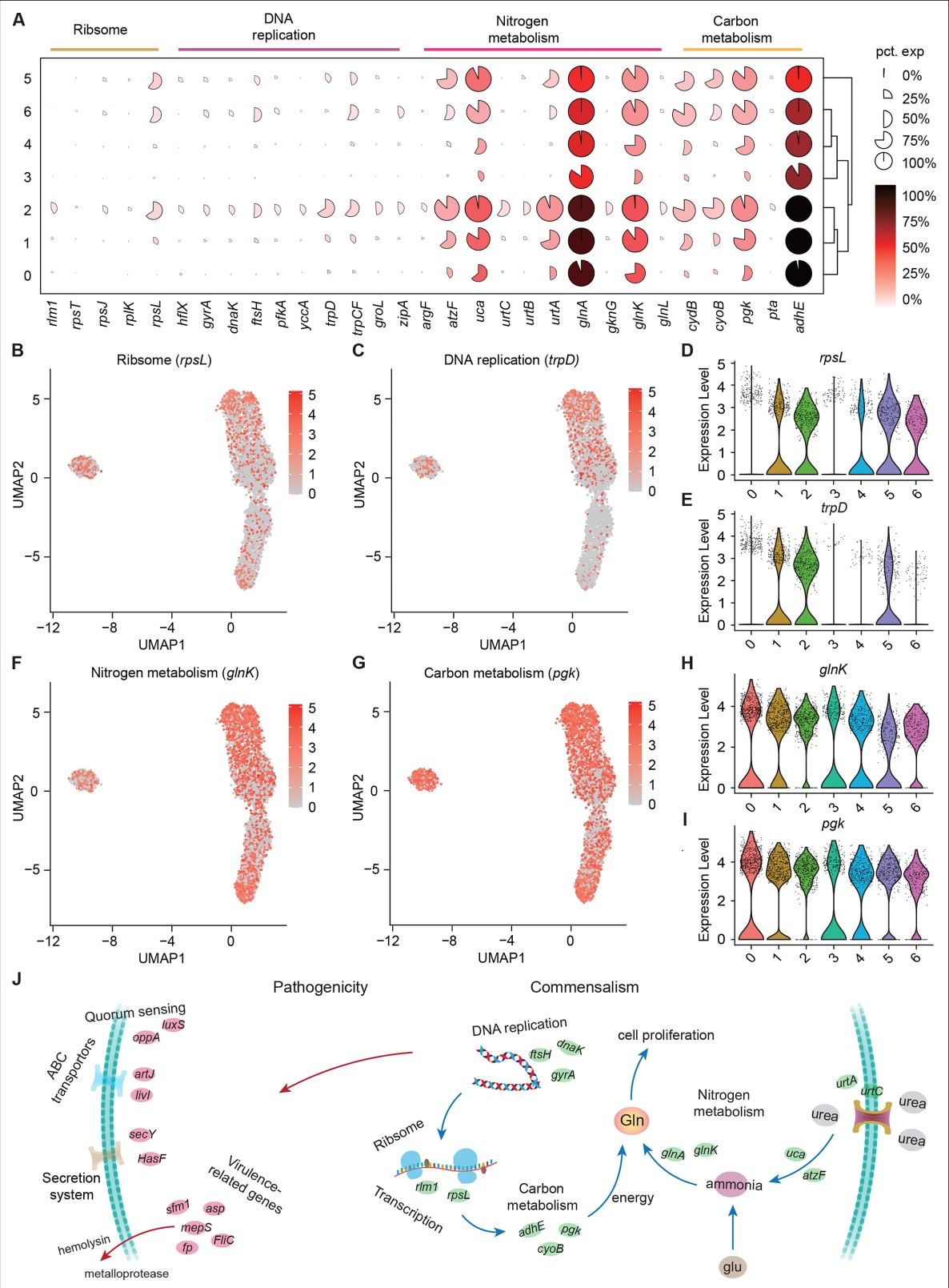

**Figure 6.** Growth heterogeneity of *S. marcescens*. (**A**) Mean expression levels of genes involved in ribsome, DNA replication, nitrogen metabolism, and carbon metabolism in different subclusters. The shape of each dot indicates the proportion of cells in the cluster, while the color indicates the average activity normalized from 0% to 100% across all clusters. (**B, C**) The expression of a representative gene of ribsome and DNA replication was highlighted on the UMAP. The red color bars represent the normalized expression of a gene across all cells analyzed. (**D, E**) Violin plots of *rpsL* and *trpD* genes in

*Figure 6 continued on next page*

*Figure 6 continued*

different subclusters. Each dot represents a single cell and the shapes represent the expression distribution. (**F, G**) The expression of two representative genes of nitrogen metabolism and carbon metabolism was highlighted on the UMAP. The red color bars represent the normalized expression of a gene across all cells analyzed. (**H, I**) Violin plots of *glnK, glnA, pgk,* and *adhE* genes in different subclusters. Each dot represents a single cell and the shapes represent the expression distribution. (**J**) Schematic of the pathogenicity and commensalism regulatory pathway.

The online version of this article includes the following figure supplement(s) for figure 6:

**Figure supplement 1.** Larvae affect the metabolism of *S. marcescens*.

## Larvae-derived AMPs efficiently antagonize *S. marcescens*

Our results showed that the host keeps strict control over the pathobiont, we next asked whether one or multiple compound(s) secreted by the host could recapitulate this phenomenon. Indeed, the data showed that diets with intestinal excreta exhibited a modest but significant decline in color intensity of the slick, prodigiosin yield, and population size compared to control (*Figure 7A–C*), in line with the result above (*Figure 2—figure supplement 2C and D*). Larval excreta contains short antimicrobial peptides (AMPs) that efficiently combat a variety of pathogens (*Marra et al., 2021*), prompting us to whether AMPs could recapitulate the response of *S. marcescens* to the presence of *Drosophila* larvae. To refine this concept, AMPs were supplemented to fly food vials with $10^7$ CFU *S. marcescens* at the same time. We found that AMPs suppressed color intensity of the slick, prodigiosin yield, and population size in a dosage-dependent manner, indicating that AMPs, to a lesser extent, played a role in reshaping *S. marcescens*. To verify it, we turned to a compound mutant strain lacking *Defensin*, *Cecropins* (four genes), *Drosocin*, *Diptericins* (two genes), *Attacins* (four genes), *Metchnikowin*, and *Drosomycin*, referred to as '*ΔAMP*' (*Hanson et al., 2019*), allowing direct investigation of their role in the lifestyle transition of the pathobiont. Indeed, *ΔAMP* recapitulated an increment in color intensity of surface slick, prodigiosin yield, and population size compared to wild-type counterparts (*Figure 7D–F*). More importantly, the recapitulation in color intensity of slick, prodigiosin yield, and population size was substantially attenuated and even abolished by the addition of AMPs, suggesting that much of the inhibition of the microbiome can be ascribed to AMPs. To rule out the potential role of other immune effectors, we turned to the IMD pathway mutant *Rel^E20* that is deficient in total immune effectors. Our result showed that the optical density and yield of prodigiosin in *Rel^E20* group did not significantly differ from the ones in *ΔAMP* group (*Figure 7—figure supplement 1A and B*). Moreover, the load of *S. marcescens* associated with *Rel^E20* mutant was comparable to that of *S. marcescens* associated with Delta AMP mutant (*Figure 7—figure supplement 1C*). Subsequently, qPCR was employed to confirm the expression of altered genes as depicted above (*Figure 7G and H*). Together, our data demonstrated that larva-derived AMPs were a key factor that accounts for the restrictive control over diet microbes.

## Discussion

The important role of the host in affecting microbial physiology and behavior is only beginning to be appreciated. In the current study, we found that *Drosophila* larvae act as a competitive regulator that prevents *S. marcescens* overgrowth and antagonizes its pathogenicity. *Drosophila* larvae reshaped the transcriptomic and metabolomic profile of *S. marcescens*, characterized by the lifestyle switch from pathogenicity to mutualism toward the fly. More importantly, we highlight that the host alters the single-cell transcriptomic atlas of *S. marcescens* and phenotypic heterogeneity in bacterial populations.

*Drosophila* adults are routinely attracted to oviposit their eggs on rotting fruits that possess both commensal and pathogenic microbes (*Liu et al., 2017*; *Wong et al., 2017*). Extensive attention has been dedicated to the essential roles of symbiotic bacteria in modulating the physiology of their animal partner, or to the molecular mechanisms underlying physiological benefits to their host. However, it's necessary to explore both sides of symbioses to obtain a more complete understanding of the host-bacteria interaction. *S. marcescens* encounters a cost associated with symbiosis, as the population size for it is diminished in the shared niche before 72 hr post inoculation by the larvae (*Figure 2D*). Our findings highlight that larvae alleviate intraspecies competition of *S. marcescens* through population size control at the initial stage. Taking into account these findings, the competition model is more plausible to the larvae-pathogen system where larvae efficiently prevent the

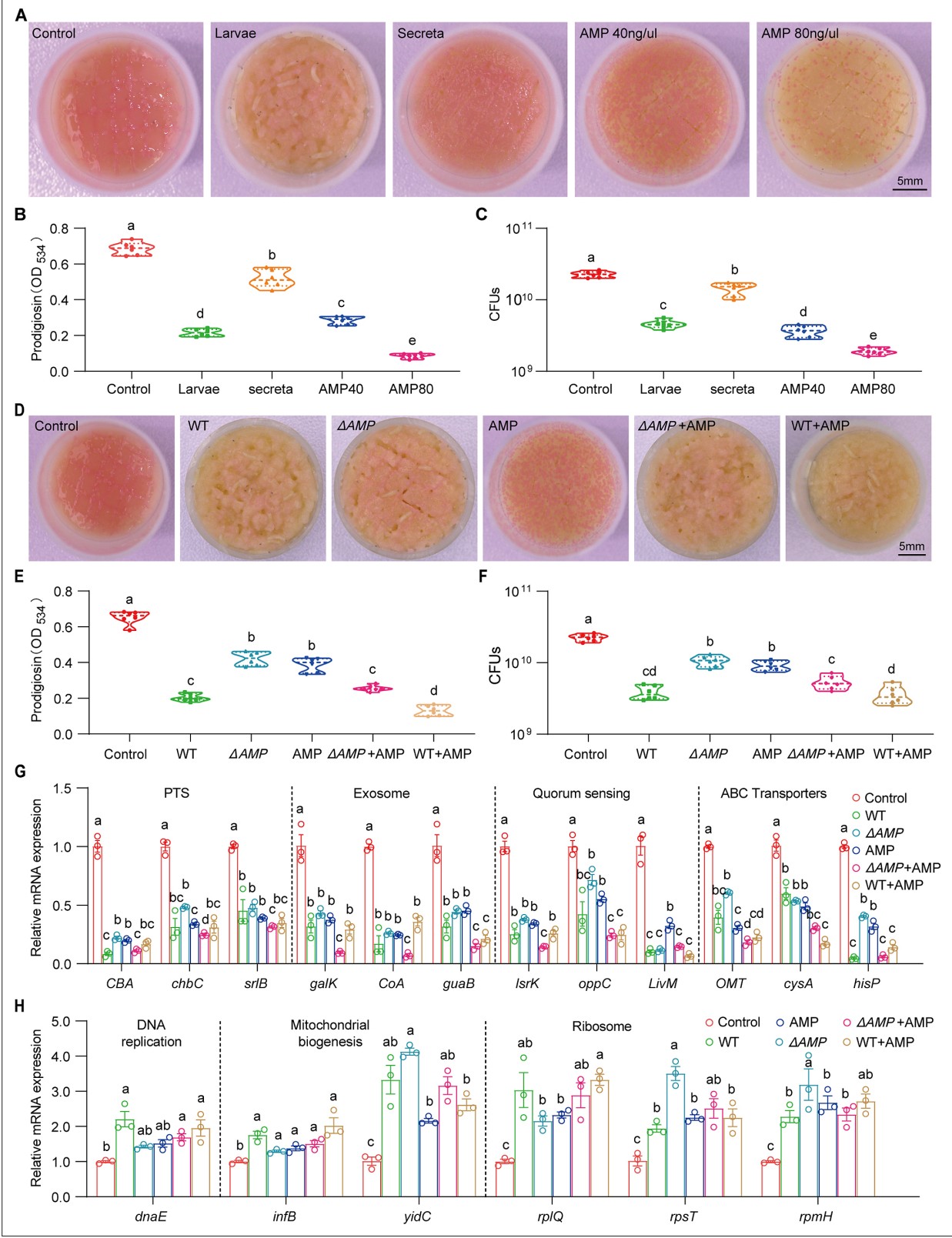

**Figure 7.** Larvae-derived antimicrobial peptides (AMPs) antagonize *S. marcescens*. (**A**) Representative images of surface slick with *S. marcescens* alone, with larvae, with secreta, and with AMPs. (**B**) The prodigiosin production of *S. marcescens* alone, with larvae, with secreta, and with AMPs. (**C**) Bacterial loads of *S. marcescens* alone, with larvae, with secreta, and with AMPs. AMPs: cecropin A (40 μg/μl, 80 μg/μl). (**D**) Representative images of surface slick with *S. marcescens* alone, with wild-type larvae, with *ΔAMP14* larvae, with AMPs, with *ΔAMP14* larvae+AMPs, and wild-type larvae+AMPs. (**E**) The

*Figure 7 continued on next page*

*Figure 7 continued*

prodigiosin production of *S. marcescens* alone, with wild-type larvae, with *ΔAMP14* larvae, with AMPs, with *ΔAMP14* larvae+AMPs, and wild-type larvae+AMPs. (**F**) Bacterial loads of *S. marcescens* alone, with wild-type larvae, with *ΔAMP14* larvae, with AMPs, with *ΔAMP14* larvae+AMPs, and wild-type larvae+AMPs. AMPs: cecropin A=40 µg/µl. (**G, H**) RT-qPCR analysis of the expression levels of downregulated and upregulated genes in the *S. marcescens* alone, with wild-type larvae, with *ΔAMP14* larvae, with AMPs, with *ΔAMP14* larvae+AMPs, and wild-type larvae+AMPs. For B–C and E–F, n=6 for each. For G–H, n=3 for each. Means ± SEMs. All variables have different letters, and they are significantly different (p<0.05). If two variables share a letter, they are not significantly different (p>0.05). Kruskal-Wallis test followed by Dunn's multiple comparisons test.

The online version of this article includes the following figure supplement(s) for figure 7:

**Figure supplement 1.** Antimicrobial peptides (AMPs) play a major role in recapitulating the response of *S. marcescens* to larvae.

overgrowth of the potential pathogenic bacterial community (**Burns et al., 2017**; **Foster et al., 2017**). Interspecies competition for nearly the same or similar nutrients and space occurs in the habitat, making *Drosophila* larvae a potential competitor for *S. marcescens*. Intriguingly, a previous study reported the contrasting result that *Drosophila* farms its commensal *L. plantarum*, and is required to optimize the extraction of dietary nutrients and sustain the growth of their symbionts upon chronic undernutrition (**Storelli et al., 2018**). To reconcile the two contradictions, another model of species cooperation is suggested. Consequently, larval transplantations favored long-term maintenance of *L. plantarum* on the diet, and benefits from symbiosis override the costs of the initial competition in the long run, conferring the beneficial effect of larval presence on bacterial maintenance. This paradox possibly reflects a strategy developed by *Drosophila* to preserve its own fitness. Potential pathogens differ from commensal bacteria, because they generate secondary metabolites threatening insect life at the static stage of growth (**Figure 2E**). If the larvae obtain benefits to promote their development and growth, they must exert strict control over them through AMPs and reactive oxygen species (**Sharp and Foster, 2022**). We found that larvae-derived AMPs efficiently antagonized *S. marcescens* (**Figure 7A–F**), thereby recapitulating the response of *S. marcescens* to *Drosophila* larvae. Yet, the full complement of larva-derived factors required for bacterial controls as well as their mode of functions remains elusive, paving the way to seek the evolutionary-conserved animal factors essential to maintain symbiosis. Taken together, our findings of the host on microbial communities would improve our understanding of the ecology of host-symbiont interactions.

Owing to their saprophagous foraging behavior, *Drosophila* has to cope with many potential pathogens in the environment (**Black et al., 2018**). We're appreciating the profound influence of the host on the resident microbial community, but the underlying molecular mechanisms by which the host is potentially involved in the perpetuation of host-bacteria symbiosis are still poorly understood. Our result showed that *S. marcescens* versatilely displayed transcriptional and metabolomic adaptations to *Drosophila* larvae (**Figure 3** and **Figure 4**), avoiding being competitors when they are associated with the host. Indeed, *S. marcescens* rapidly reached the plateau and exhausted its nutritional resources, and then generated secondary metabolites that could endanger insect life (**Figure 2E**). Consistently, our transcriptome data suggest the pathogen-to-commensal transition of *S. marcescens* by the presence of larvae, including ABC transporters, phosphotransferase system, quorum sensing, and exosome. In gram-negative bacteria, these exporters transport lipids and polysaccharides from the cytoplasm to the periplasm, or certain substances that need to be extruded from the cell, including surface components of the bacterial cell (e.g. capsular polysaccharides, lipopolysaccharides, and teichoic acid), proteins involved in bacterial pathogenesis (e.g. hemolysis), hydrolytic enzymes, competence factors, toxins, bacteriocins, peptide antibiotics, and siderophores. In addition, quorum sensing, used by pathogens in disease and infection, regulates gene expression following population density through autoinducers, allowing bacterial populations to communicate and coordinate group behavior (**Mukherjee and Bassler, 2019**). The findings suggest that the microbiome responds to host physiology by altering gene expression and metabolism (**Penterman et al., 2014**). Consistently, a study observed that commensal bacteria calibrate their transcriptional and metabolic output to different systemic inflammatory responses (**Fyhrquist et al., 2019**). Taken together, these findings suggest that larvae elicit a set of expressed genes in their harmful symbionts, favoring the shift from a pathogenic to a commensal stage.

Most microbial communities consist of a genetically diverse assembly of different organisms, and the level of genetic diversity plays an important part in community properties and functions (**Davies et al., 2022**). However, genetically identical microbial cells can show different behaviors, including

differences in growth speed, gene expression, and metabolism, and biological diversity can arise at a lower level of biological units (*Ackermann, 2015*; *Avraham et al., 2015*). Fortunately, the recent development of methods to interrogate populations on the single-cell level has dramatically altered our understanding of cellular heterogeneity by providing much greater resolution of different cell types and cell states (*Hare et al., 2021*; *Imdahl et al., 2020*; *McNulty et al., 2023*). By applying this technique, we indeed observed that single cells differ from each other with respect to gene expression even when genetic and environmental differences between cells are reduced as much as possible in a shaking liquid medium (*Figure 5—figure supplement 1*). Recent development in single-cell techniques facilitates to reveal that distinct bacterial subpopulations contribute unique colonization and growth strategies to infection sites (*Lloréns-Rico et al., 2022*). For example, the host can drive *Salmonella* phenotypic heterogeneity at the single-cell level throughout the course of infection, highlighting how variation in gene expression and metabolic activity contribute to the overall bacterial success (*Tsai and Coombes, 2019*). Consistently, we found that the host changes the transcriptomic atlas of *S. marcescens* individual cells, and attenuates phenotypic heterogeneity of virulent factors (*Figure 5D–L*). To date, transcriptional approaches – profiling the host, the pathogen, or both – have been employed to uncover substantial molecular details about the host and bacterial factors that underlie infection outcomes. During the interaction, the niche environment diversifies into 3D areas with varying degrees of growth conditions (*Koo and Yamada, 2016*). In a single population, both in vitro and in vivo, *Salmonella typhimurium* has been shown to display significant cell-to-cell variation in attributes such as growth rate, expression of virulence factors, and sensitivity to antibiotics (*Claudi et al., 2014*). With the development of this approach, it is possible to identify and modify the transcription of certain target bacterial cells with the expression of virulence factors in order to selectively treat human diseases in the gut microenvironment.

Utilizing the *Drosophila* model system, we revealed a natural ecological phenomenon whereby the host was a prerequisite for regulating the population size and lifecycle switch of indigenous bacteria. It's of importance to understand the ecological and evolutionary processes that shape host-associated microbial communities. Little is known about the effect of the host on the commensal bacteria, and it would be also interesting to investigate whether the host and commensal bacteria could synergize to antagonize the pathogenicity of potential pathogens. Future studies that evaluate the molecular mechanism underlying the effect of the host on microbial communities would improve our understanding of host-symbiont co-evolution in nature.

# Materials and methods

**Key resources table**

| Reagent type (species) or resource | Designation | Source or reference | Identifiers | Additional information |
|---|---|---|---|---|
| Strain, strain background (*Drosophila melanogaster*) | *Canton S* | This laboratory | N/A | |
| Strain, strain background (*Drosophila melanogaster*) | *yw; Sp/CyO; MKRS/TM6B* | *Liu et al., 2012* | N/A | |
| Strain, strain background (*Drosophila melanogaster*) | *dfmr1*$^{50M}$ | *Liu et al., 2012* | N/A | |
| Strain, strain background (*Drosophila melanogaster*) | *ΔAMP14* | *Marra et al., 2021* | N/A | |
| Strain, strain background (*Drosophila melanogaster*) | *Rel*$^{E20}$ | *Marra et al., 2021* | N/A | |
| Strain, strain background (*Serratia marcescens*) | *S. marcescens FY* | *Liu et al., 2020* | GenBank: CP053378 | |
| Strain, strain background (*Lactobacillus plantarum*) | *L.P* | *Liu et al., 2020* | GenBank: KY038178 | |
| Peptide, recombinant protein | Cecropin A | Sigma-Aldrich | C6830 | |
| Commercial assay or kit | QIAamp Fast DNA stool Mini Kit | QIAGEN | 52504 | |

*Continued on next page*

*Continued*

| Reagent type (species) or resource | Designation | Source or reference | Identifiers | Additional information |
|---|---|---|---|---|
| Commercial assay or kit | Bacteria RNA Extraction Kit | Vazyme | R403 | |
| Commercial assay or kit | Ribo-off rRNA Depletion Kit | Vazyme | N407 | |
| Commercial assay or kit | VAHTS Universal V8 RNA-seq Library Prep Kit for Illumina | Vazyme | NR605 | |
| Commercial assay or kit | VAHTS DNA Clean Beads | Vazyme | N411 | |
| Commercial assay or kit | VAHTS RNA Clean Beads | Vazyme | N412 | |
| Commercial assay or kit | Murine RNase inhibitor | Vazyme | R301 | |
| Commercial assay or kit | HiScript III All-in-one RT SuperMix Kit | Vazyme | R333 | |
| Commercial assay or kit | ChamQ Universal SYBR qPCR master mix kit | Vazyme | Q711 | |
| Chemical compound, drug | Sodium benzoate | Sigma-Aldrich | 532-32-1 | |
| Chemical compound, drug | Sodium hypochloride | Sigma-Aldrich | 239305 | |
| Chemical compound, drug | Formaldehyde | Sigma-Aldrich | 47608 | |
| Chemical compound, drug | Kanamycin | Sigma-Aldrich | 420411 | |
| Chemical compound, drug | Ampicillin | Sigma-Aldrich | A5354 | |
| Chemical compound, drug | Serotonin | Sigma-Aldrich | 14927 | |
| Chemical compound, drug | Sorbitol | Sigma-Aldrich | 240850 | |
| Chemical compound, drug | Lactobionic acid | Macklin | 96-82-2 | |
| Sequence-based reagent | Primers for real-time qPCR: **Supplementary file 1** | This paper | N/A | |
| Software, algorithm | Image Processing | **Schneider et al., 2012** | https://imagej.net/software/fiji/ | |
| Software, algorithm | GraphPad Prism 9.0 | GraphPad Software | https://www.graphpad.com/ | |
| Software, algorithm | Adobe Illustrator 2023 | Adobe | https://www.adobe.com/ | |
| Software, algorithm | DESeq 2 | DESeq | http://bioconductor.org/packages//2.10/bioc/html/DESeq.html | |
| Software, algorithm | MZmine 2.53 | MZmine 2 | http://mzmine.github.io/ | |
| Software, algorithm | MultiQuant | Sciex | https://www.sciex.com/ | |
| Software, algorithm | StarSolo 2.7.10a | **Dobin, 2024** | https://github.com/alexdobin/STAR | |
| Software, algorithm | R Version 3.5.1 | **R Development Core Team, 2018** | https://www.r-project.org/ | |
| Software, algorithm | Seurat 4.3.0.1 | Seurat | N/A | |
| Other | Bulk RNA-seq data | This paper | GEO: GSE232120 | Deposited data |
| Other | Single-cell RNA-seq data | This paper | GEO: GSE232484 | Deposited data |
| Other | Metabolomics data | This paper | MTBLS7962 | Deposited data |

## Fly culture and stocks

*Drosophila* flies were routinely reared and kept at the condition of 25°C, 55–65% humidity with a 12 hr:12 hr light-dark circle unless otherwise noted. The Canton S strain was used as the wild-type fly in this work. Delta *14 AMP* mutant and *Rel^{E20}* mutant was generated as described (**Hanson et al., 2019**) and kindly gifted by Dr. Zhai in Hunan Normal University. The *yw; Sp/CyO; MKRS/TM6B* and *dfmr1^{50M}* null mutant were used as weak flies and infertile flies (**Liu et al., 2012**). The fly was raised on

standard cornmeal-sugar-agar medium (1 l): 105 g dextrose, 7.5 g agar, 26 g yeast, 50 g cornmeal, 0.25 g sodium benzoate (Sigma-Aldrich) dissolved in 8.5 ml 95% ethanol and 1.9 ml propionic acid (99%, Mallinckrodt Baker).

## Generation of GF and gnotobiotic flies

GF flies were generated as described (*Jia et al., 2021*). In brief, fresh embryos within 8 hr post egg-laying were collected from agar media with 1.5% grape juice, rinsed with ddH$_2$O, and transferred into 1.5 ml Eppendorf tubes. Diluted sanitizer Walch (1:30), 2.5% sodium hypochloride (Sigma-Aldrich), 75% ethanol, and sterile PBS containing 0.01% Triton X-100 were successively applied to bleach embryos. The GF embryos were cultivated in autoclaved fly food with 10% yeast. Axenia of GF larvae was normally tested by plating the larval homogenates on nutrient agar plates (peptone 10 g/l; beef extract powder 3 g/l; NaCl 5 g/l; agar 15 g/l). Gnotobiotic flies were generated by inoculating bacterial strains to GF flies.

## Survival assay

Newly eclosed female flies were collected and transferred into vials (15 flies/vial), and each group contained 6 vials with a total of 180 flies in each group based on two replicates for each. Before bacterial treatment, vials were pre-inoculated with *S. marcescens* alone and *S. marcescens* with 40 larvae, and incubated at 25°C for 24 hr for culture. The flies were then transferred to vials processed with *S. marcescens* alone and *S. marcescens* in-coculture every 3 days. The number of dead flies was counted each day, and the proportion of surviving flies was calculated at each timepoint of the experiment. Experiments were independently replicated twice.

## Bacterial culture and bacterial load quantification

All the material to manipulate bacteria was sterilized before usage. The strains of *S. marcescens* with the GenBank accession number CP053378 and *L. plantarum* with the GenBank accession number KY038178 were used. *S. marcescens* and *L. plantarum* were cultured in LB and MRS broth medium at 30°C, respectively. Bacterial cells were harvested by centrifugation (5000 rpm, 3 min), washed twice in 1× PBS, and resuspended in 1× PBS to obtain $10^8$ cells/ml (OD$_{595}$=1). The bacterial suspension (20 µl) was supplemented to a vial with autoclaved fly food, and crawling GF larvae were then added to vials to generate a host-microbe interaction model. For the heat-shock experiment, the culture was transferred to a 45°C incubator at 12 hr timepoint after post-inoculation, and kept for 15 min. Heat-shocked bacteria were used for bacterial single-cell RNA-seq below.

Fly food in vials was agitated and homogenated with 5 ml ddH$_2$O. Bacterial load was assessed by plating 10-fold serial dilutions of the food homogenates on LB agar plates and incubating the plates at 30°C for 16 hr. Mixed bacteria (bacteria in the living environment of *Drosophila*) were quantified in the NA medium that supports the growth of *Drosophila* microbiota (*Jia et al., 2021*). The numbers of CFUs were counted, and expressed as the total number of living bacteria per vial.

## Prodigiosin production assay

The determination of the prodigiosin yield of bacteria was carried out with acidified ethanol and absorbance measurement as previously described (*Kalivoda et al., 2010*; *Pan et al., 2020*). The relative concentration of prodigiosin produced by solid-grown cultures was quantified as follows. Samples were added to 1.2 ml acidified ethanol (4% 1 M HCl in ethanol) to extract prodigiosin from the culture for 10 min. Cell debris and impurities were removed by centrifugation at 13,000 rpm for 5 min, and the supernatant was transferred to a cuvette for measurement of absorbance at 534 nm. Prodigiosin production of samples was calculated as the optical density at 534 nm. Experiments were independently replicated four times.

## AMP and antibiotics treatment

Cecropin A produced by silkworms (Sigma-Aldrich) was used as the representative of AMPs. Cecropin A was added to food at 40 µg/µl and 80 µg/µl. Antibiotic food was prepared by adding 5 µg/µl kanamycin (Sigma-Aldrich) and 10 µg/µl ampicillin (Sigma-Aldrich). The prodigiosin concentration and bacterial loads of *S. marcescens* challenged with cecropin A, kanamycin, and ampicillin were examined as described above.

## Transmission electron microscopy

TEM was conducted to observe the structures of the bacterial cells as previously depicted (*Mörgelin, 2017*). Briefly, bacterial cells were collected after 24 hr incubation and then fixed in 2.5% glutaraldehyde for 5 hr. The cells were dehydrated in a gradient series of ethanol solutions from 30% to 100% by incubation. For TEM analysis, the samples were treated with acetone and embedded in epoxy resin. Thin sections (70 nm) were cut with a diamond knife mounted on a Leica UC-7 ultramicrotome and collected on carbon-coated Cu grids. TEM observation was performed with a JEOL JEM1400 TEM at 200 kV. Micrograph films were digitally acquired at high resolution with EMSIS Morada G3 (*Winey et al., 2014*).

## Real-time PCR analysis

RT-qPCR assay was performed as described previously (*Liu et al., 2022*). In brief, to assess the expression levels of prodigiosin synthesis-related genes, extracellular polysaccharide production-related genes, and partial downregulated and upregulated genes, the cultures of different groups of *S. marcescens* were collected after 24 hr incubation. The collected cells were then subjected to total RNA extraction using a Bacteria RNA Extraction Kit (Vazyme, China). After treating the total bacterial RNA with DNase I (Vazyme, China), the concentration and quantity of the total bacterial RNA were determined using a NanoDrop spectrophotometer (Thermo Scientific), and 0.6 µg of the total bacterial RNA was subjected to reverse transcription using the HiScript III All-in-one RT SuperMix Kit (Vazyme). The mixture was subjected to RT-qPCR analysis using the ChamQ Universal SYBR qPCR master mix kit (Vazyme) in a CFX96 Real-Time System (Bio-Rad, Hercules, CA, USA). RNA from three biological replicates were analyzed and four technical replicates were performed. The relative expression values were calculated using the following formula: $\Delta Ct = Ct$ (target gene) - $Ct$ (reference gene), and the relative expression was equal to $2^{-\Delta Ct}$. The 16S rRNA protein-encoding gene was used as an internal control. The primers used for RT-qPCR analysis are listed in *Supplementary file 1*. Experiments were independently replicated three times.

## Bacterial bulk RNA-seq

To analyze the effect of larvae on the transcriptome in *S. marcescens*, the cultures of different groups were collected after 24 hr incubation. Ice-cold 1× PBS (2 ml) was added to the vial for 5 min to suspend bacterial cells, and then suspended bacterial cells were removed food remainder by centrifugation (900 rpm, 3 min), washed twice in 1× PBS. Bacterial cells were harvested by centrifugation (5000 rpm, 4 min) and washed twice in 1× PBS. The collected cells were then subjected to total RNA extraction using a Bacteria RNA Extraction Kit (Vazyme, China). The integrity, concentration, and quantity of the bacterial RNA were determined using a NanoDrop spectrophotometer (Thermo Scientific) and a 1% agarose gel. 1 µg total RNA with RIN value above 7 was used for the following strand-specific library construction using VAHTS Universal V8 RNA-seq Library Prep Kit for Illumina (Vazyme, China). Library quality was evaluated using an Agilent 2100 Bioanalyzer. The sequences were sequenced and processed using Novaseq-PE150 by the company (Novogene, Beijing, China).

For annotation, the genome of *S. marcescens FY* was used as a reference. The significant DEGs were determined in different groups using the DESeq software (DESeq 2), with the standards of p-value≤0.05, and fold change |log2Ratio|≥1.2. KEGG KAAS database was used to annotate the genes with significantly differential expression based on their functions using the BLAST.

## Metabolomics analysis

Untargeted metabolomics was performed using a modified version of a previously reported protocol (*Tsugawa et al., 2015*). In brief, L-2-chlorophenylalanine (0.06 mg/ml) dissolved in methanol was taken as internal standard, the samples were performed to GC-MS analysis (Shanghai OE Biotech Co., Ltd). QC samples were prepared by mixing aliquots of all samples to be a pooled sample. Samples were analyzed on an Agilent 7890B gas chromatography system coupled to an Agilent 5977A MSD system (Agilent Technologies Inc, CA, USA). The temperature of the MS quadrupole and ion source (electron impact) was set to 150°C and 230°C, respectively. The collision energy was 70 eV. Mass spectrometric data were acquired in a full-scan mode (m/z 50–500). For data processing, MZmine 2 and MultiQuant software programs were used.

## Bacterial single-cell RNA-seq and analysis

Bacterial single-cell RNA-seq was carried out on a commercially available platform (M20 Genomics Company) (*Xu et al., 2023*). In brief, half a million *S. marcescens* cells were collected by centrifuging at 4°C. The supernatant was removed and cell pellets were resuspended with 2 ml ice-cold 4% formaldehyde (Sigma, 47608). The samples were incubated with shaking overnight at 4°C. The fixative was removed by centrifuging at 4000 rpm for 5 min at 4°C and cells were washed twice with 1 ml PBS-TRI (1× PBS supplemented with 0.1% Tween-20 and 0.1 U/ml Murine RNase inhibitor [Vazyme, R301]). The supernatant was removed and cells were resuspended in 200 µl PBS-TRI (1× PBS supplemented with 0.1% Tween-20 and 0.2 U/ml Murine RNase inhibitor [Vazyme, R 301]). The cell suspensions were counted with a Moxi cell counter and diluted according to the manufacturer's instructions to obtain single cell.

The bacterial single-cell RNA-seq library was prepared according to the protocol of VITAPilote kit (M20 Genomics, R20114124). In situ reverse transcription of bacteria was performed with random primers and the resulting cDNA fragment was added with adaptor. The droplet barcoding for a single bacterium was performed on VITACruiser Single Cell Partitioning System (M20 Genomics, Hangzhou, China). Bacteria, DNA extension reaction mix, and hydrogel barcoded beads were encapsulated using the VITACruiser. The aqueous phase containing cDNAs was purified with magnetic beads. The cDNAs were amplified by PCR and purified with magnetic beads. All products were pooled to construct a standard sequencing library. Sequencing was done on a PE150 (Illumina), and raw reads were aligned against the *S. marcescens FY* genome and counted by StarSolo followed by secondary analysis in the Annotated Data Format. Sequencing data was further analyzed using Seurat (v.4.3.0.1 with default parameters except where indicated). Cells were filtered to retain only those with at least 100 genes. Genes were also screened to remove genes expressed only in fewer than five cells. Then, we first log-transformed the data using the NormalizeData function, and selected the 2000 most variable genes using FindVariableFeatures. Then, we z-scored these highly variable genes using 'ScaleData'. Next, we performed linear dimensionality reduction using PCA down to 50 dimensions ('RunPCA'). Points in this embedding were used to construct UMAP plots and find neighbors for clustering. FindClusters was used to run the Louvain clustering algorithm and generate clusters. We confirmed that the clusters highlighted in the main text appeared consistently for a range of resolutions from 0.5 to 1.5. DEG was performed using the FindMarkers function with a log-fold change cutoff of 0.25.

## Statistical analysis

Statistical analysis is performed using GraphPad Prism 9.0 and indicated inside each figure legend. The layout of all figures used Adobe Illustrator 2022. Experimental flies were tested at the same condition, and all data are collected from at least two independent experiments. D'Agostino-Pearson normality test was used to verify the normal distribution of data. If normally distributed, a two-tailed Student's t-test was used to compare two groups, and one-way ANOVA with two-stage step-up method of Benjamini, Krieger, and Yekutieli followed by Tukey's multiple comparisons test was used for comparisons of multiple groups. If not normally distributed, a two-tailed Mann-Whitney U-test was performed to compare two groups of samples, while Kruskal-Wallis test followed by Dunn's multiple comparisons test was used for multiple comparisons among three or more groups.

## Acknowledgements

The authors would like to thank all members of Dr. Liu's and Dr. Wang's labs for discussion. We appreciate S Wang, G Wang, Y Pan, Z Zhai, and Y Zhang for their critical comments on the manuscript. We thank the Bloomington Stock Center for providing fly stocks and Dr. Zongzhao Zhai for fly stocks. This work was supported by the National Natural Science Foundation of China (31501175), Natural Science Foundation of Anhui Province (2308085MC74), the Foundation of Anhui Province Key Laboratory of Resource Insect Biology and Innovative Utilization (FKLRIB202401) and Talents in Anhui Agricultural University (RC342201) to WL, and the Ministry of Science and Technology of the People's Republic of China (2022YFE0132000), the Natural Science Foundation of Hunan Province (2022JJ40048), and the Fundamental Research Funds for the Central Universities of China (531118010546) to YW. The funders had no role in study design, data collection, and interpretation, or the decision to submit the work for publication.

## Additional information

### Funding

| Funder | Grant reference number | Author |
|---|---|---|
| National Natural Science Foundation of China | 32470044 | Wei Liu |
| Natural Science Foundation of Anhui Province | 2308085MC74 | Wei Liu |
| Anhui Province Key Laboratory of Resource Insect Biology and Innovative Utilization | FKLRIB202401 | Wei Liu |
| Talents in Anhui Agricultural University | RC342201 | Wei Liu |
| Ministry of Science and Technology of the People's Republic of China | 2022YFE0132000 | Yirong Wang |
| Natural Science Foundation of Hunan Province | 2022JJ40048 | Yirong Wang |
| Fundamental Research Funds for the Central Universities of China | 531118010546 | Yirong Wang |
| National Natural Science Foundation of China | 31501175 | Wei Liu |

The funders had no role in study design, data collection and interpretation, or the decision to submit the work for publication.

### Author contributions

Ziguang Wang, Conceptualization, Data curation, Formal analysis, Writing - original draft; Shuai Li, Sheng Zhang, Software, Formal analysis, Visualization, Methodology; Tianyu Zhang, Methodology; Yujie Wu, Data curation, Investigation; Anqi Liu, Validation, Investigation; Kui Wang, Methodology, Project administration; Xiaowen Ji, Supervision, Methodology, Project administration; Haiqun Cao, Supervision, Project administration; Yinglao Zhang, Supervision, Methodology; Eng King Tan, Project administration, Writing – review and editing; Yongcheng Wang, Methodology, Project administration, Writing – review and editing; Yirong Wang, Resources, Methodology, Project administration, Writing – review and editing; Wei Liu, Resources, Funding acquisition, Methodology, Project administration, Writing – review and editing

### Author ORCIDs

Yirong Wang https://orcid.org/0000-0002-1995-2728
Wei Liu https://orcid.org/0000-0002-1059-6261

Reviewer #1 (Public Review): https://doi.org/10.7554/eLife.96789.3.sa1
Reviewer #3 (Public Review): https://doi.org/10.7554/eLife.96789.3.sa2
Author response https://doi.org/10.7554/eLife.96789.3.sa3

---

## Additional files

### Supplementary files

• MDAR checklist

• Supplementary file 1. Primers used for PCR experiments.

• Supplementary file 2. The list of genes with a significant change in bulk RNA-seq.

• Supplementary file 3. The list of metabolites with a signifcant change in metabolomics.

• Supplementary file 4. The single cell transcriptional profile of pathogenicity and bacterial propagation.

### Data availability

Sequencing data have been deposited in GEO under accession codes GSE232120 and GSE232484. Metabolomics data have been deposited in MTBLS7962. All data generated or analysed during this study are included in the manuscript and supporting files.

The following datasets were generated:

| Author(s) | Year | Dataset title | Dataset URL | Database and Identifier |
| --- | --- | --- | --- | --- |
| Wang Z, Li S, Wang Y, Liu W | 2024 | The Host Modulates Transcriptional Profile and Phenotypic Heterogeneity in Symbionts | https://www.ncbi.nlm.nih.gov/geo/query/acc.cgi?acc=GSE232120 | NCBI Gene Expression Omnibus, GSE232120 |
| Wang Z, Li S, Wang Y, Liu W | 2024 | The Host Modulates Transcriptional Profile and Phenotypic Heterogeneity in Symbionts | https://www.ncbi.nlm.nih.gov/geo/query/acc.cgi?acc=GSE232484 | NCBI Gene Expression Omnibus, GSE232484 |

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
