## [Editor Report · eLife assessment]

The **fundamental** findings of this work substantially advance our understanding of the impact of the host on its gut microbes. The authors provided **compelling** evidence at single-cell resolution that the host can drive heterogeneity in the populations of gut microbes with significant consequences for the host physiology.

---

## [Referee Report · Reviewer #1 (Public Review)]

Summary:

In this work, Wang and colleagues used *Drosophila*-Serratia as a host-microbe model to investigate the impact of the host on gut bacteria. The authors showed that *Drosophila* larvae reduce S. marcescens abundance in the food likely due to a combination of mechanical force and secretion of antimicrobial peptides. S. marcescens exposed to *Drosophila* larvae lost virulence to flies and could promote larval growth similar to typical *Drosophila* gut commensals. These phenotypic changes were reflected in the transcriptome and metabolome of bacteria, suggesting that the host could drive the switch from pathogenicity to commensalism in bacteria. Further, the authors used single-cell bacterial RNA-seq to demonstrate the heterogeneity in gut bacterial populations.

Strengths:

This is a valuable work that addresses an important question of the impact of the host on its gut microbes. The authors could convincingly demonstrate that gut bacteria are strongly affected by the host with important consequences for both interacting partners. Moreover, the authors used state-of-the-art bacterial single-cell RNA-seq to reveal heterogeneity in host-associated commensal populations.

Overall most parts of the study are solid and clear.

---

## [Referee Report · Reviewer #3 (Public Review)]

In this study, Wang and coworkers established a model of *Drosophila*-S. marcescens interactions and thoroughly examined host-microbe bidirectional interactions. They found that:

(1) *Drosophila* larvae directly impact microbial aggregation and density;

(2) *Drosophila* larvae affect microbial metabolism and cell wall morphology, as evidenced by reduced prodigiosin production and EPS production, respectively;

(3) *Drosophila* larvae attenuate microbial virulence;

(4) *Drosophila* larvae modulate the global transcription of microbes for adaptation to the host;

(5) Microbial single-cell RNA sequencing (scRNA-seq) analysis revealed heterogeneity in microbial pathogenicity and growth;

(6) AMPs are key factors controlling microbial virulence phenotypes.

Taken together, they concluded that host immune factors such as AMPs are directly involved in the pathogen-to-commensal transition by altering microbial transcription.

In general, in this revised version, I feel that the authors addressed all the points raised in the previous review process. Specifically, they demonstrated that sub-lethal doses of antibiotics such as kanamycin or ampicillin is sufficient to induce the virulence switch in S. marcescens. Furthermore, by testing IMD pathway mutant animals, they concluded that AMP plays a major role in the commensal-to-pathogen transition. In summary, I appreciate the authors' efforts, and I am satisfied with the revision.

---

## [Author Response]

The following is the authors’ response to the original reviews.

**eLife assessment**
This valuable study examines the role of a host in conditions that shift pathogenicity of opportunistic microbes. The use of single-cell microbial transcriptomics and metabolomics to demonstrate the host's effects on pathogen dynamics is interesting and convincing. However, the connection to host antimicrobial peptides driving these effects is incomplete and would benefit from additional evidence and improved explanation in the text. This paper has the potential to be of broad interest to those working in host-microbe (microbiome and pathogen) interactions.

We appreciate the editors for organizing our manuscript and providing eLife assessment. We went through each comment and carried out some necessary experiments. According to the comments, we here provide additional evidence that further supports our findings in this revised manuscript.

**Public Reviews:**

**Reviewer #1 (Public Review):**
Summary:In this work, Wang and colleagues used *Drosophila*-Serratia as a host-microbe model to investigate the impact of the host on gut bacteria. The authors showed that *Drosophila* larvae reduce S. marcescens abundance in the food likely due to a combination of mechanical force and secretion of antimicrobial peptides. S. marcescens exposed to *Drosophila* larvae lost virulence to flies and could promote larval growth similar to typical *Drosophila* gut commensals. These phenotypic changes were reflected in the transcriptome and metabolome of bacteria, suggesting that the host could drive the switch from pathogenicity to commensalism in bacteria. Further, the authors used single-cell bacterial RNA-seq to demonstrate the heterogeneity in gut bacterial populations.Strengths:This is a valuable work that addresses an important question of the effect of the host on its gut microbes. The authors could convincingly demonstrate that gut bacteria are strongly affected by the host with important consequences for both interacting partners. Moreover, the authors used state-of-the-art bacterial single-cell RNA-seq to reveal heterogeneity in host-associated commensal populations.Weaknesses:Some of the conclusions are not fully supported by the data.Specifically, in lines 142-143, the authors claim that larva antagonizes the pathogenicity of S. marcescens based on the survival data. I do not fully agree with this statement. An alternative possibility could be that, since there are fewer S. marcescens in larvae-processed food, flies receive a lower pathogen load and consequently survive. Can the authors rule this out?Also, the authors propose that *Drosophila* larvae induce a transition from pathogenicity to commensalism in S. marcescens and provide nice phenotypic and transcriptomic data supporting this claim. However, is it driven only by transcriptional changes? Considering high mutation rates in bacteria, it is possible that S. marcescens during growth in the presence of larvae acquired mutations causing all the observed phenotypic and transcriptional changes. To test this possibility, the authors could check how long S. marcescens maintains the traits it acquires during growth with *Drosophila*. If these traits persist after reculturing isolated bacteria, it is very likely they are caused by genome alterations, if not - likely it is a phenotypic switch driven by transcriptional changes.

We thank the reviewer for providing a feasible method to distinguish the shift in transcriptional profile from genomic mutations. According to this valuable suggestion, we checked phenotypic and transcriptional changes after re-culturing the bacterium that had coexisted with larvae. We found that all phenotypes can be recovered after re-culturing. The new data supported our previous result that a phenotypic switch was driven by transcriptional changes rather than genome mutations. We now add these results to the text with figure supplement 3 (line 147-151, 192-194). Please see the following text.

“To rule out the possibility that phenotypic alterations could stem from genomic mutations, we examined the prodigiosin yield and CFUs of re-culturing *S. marcescens* that had coexisted with larvae. Our results showed that neither prodigiosin yield nor CFUs of re-culturing *S. marcescens* differed from the original strain (Figure 2-figure supplement 3A-C), suggesting that a phenotypic switch was driven primarily by transcriptional reprogramming.” “Consistent with the previous result that this phenotypic switch was driven by transcriptional changes, the expression of virulent and growth genes was recovered after re-culturing (Figure 3-figure supplement 3D, E).”

For the first question, we admit the possibility that the high morality of flies could result from the acquirement of a higher pathogen load, because of an increase in the bacterial load of single *S. marcescens*. However, host pathogenesis is normally determined by the virulence of pathogens rather than the number of bacteria. For example, hosts constantly harbor astonishing commensals in their guts, but remain healthy. This evidence suggests that it was the property (virulence) of a pathogen that is more important to affect the health status of the hosts. Moreover, an increase in virulence of single *S. marcescens* was verified by real-time PCR (Fig. 2F) and TE (Fig. 2G). Taken together, we could draw a conclusion that the impaired survival of flies challenged with single *S. marcescens* mainly arose from an increase in the virulence of *S. marcescens*. Thanks for your understanding!

**Reviewer #2 (Public Review):**
Summary:While many studies have explored the impacts of pathogens on hosts, the effect of hosts on pathogens has received less attention. In this manuscript, Wang et al. utilize *Drosophila melanogaster* and an opportunistic pathogen, Serratia marcescens, to explore how the host impacts pathogenicity. Beginning with an observation that larval presence and density impacted microbial growth in fly vials (which they assess qualitatively as the amount of 'slick' and quantitatively as microbial load/CFUs), the authors focus on the impact of axenic/germ-free larvae on an opportunistic pathogen S. marcescens. Similar to their observations with general microbial load, they find that larvae reduce the presence of a pinkish slick of Sm, indicative of its secondary metabolite prodigiosin. The presence of larvae alters prodigiosin production, pathogen load, pathogen cellular morphology, and virulence, and this effect is through transcriptional and metabolic changes in the pathogen. Overall, they observe a loss of virulence factors/pathways and an increase in pathways contributing to growth. Given the important role the host plays in this lifestyle shift, the authors then examined host features that might influence these effects, focusing on the role of antimicrobial peptides (Amps). The authors combine the use of synthetic Amps and an Amp-deficient fly line and conclude much of the larval inhibitory effect is due to their production of AMPs.Strengths:This is a very interesting question and the use of *Drosophila*-Serratia marcescens is a great model to explore these interactions and effects.The authors have an interesting and compelling phenotype and are asking a unique question on the impact of the host on the pathogen. The use of microbial transcriptomics and metabolomics is a strength, especially in order to assess these impacts on the pathogen level and at the single-cell level to capture heterogeneity.Weaknesses:Overall, the writing style in the manuscript makes it difficult to fully understand and appreciate the data and its interpretation.The data on the role of AMPs would benefit from strengthening. Some of the arguments in the text of that section are also counterintuitive. The authors show that △AMP larvae have a reduced impact on Sm as compared to wt larvae, but it seems less mild of an effect than that observed with wt excreta (assuming the same as secreta in Figures 7, should be corrected or harmonized). Higher doses of AMPs give a phenotype similar to wt larvae, but a lower dose (40 ng/ul) gives phenotypes more similar to controls. The authors argue that this data suggests AMPs are the factor responsible for much of the inhibition, but their data seems more to support that it's synergistic- you seem to still need larvae (or some not yet defined feature larvae make, although secreta/excreta was not sufficient) + AMPs to see similar effects as wt. Based on positioning and color scheme guessing that AMP 40ng/ul was used in Figures 7D-H, but could not find this detail in the text, methods, or figure legend and it should be indicated. This section does not seem to be well supported by the provided data, and this inconsistency greatly dampened this reviewer's enthusiasm for the paper.

We thank the reviewer’s valuable comments and suggestions. We admitted that some photos of the pinkish slick (prodigiosin) are counterintuitive in Figure 7 as well as figure supplement 2B. Here comes the reason. Single *S. marcescens* produced prodigiosin that only stayed on the surface of fly agar medium. As we know, larvae can agitate food and form a stratification of prodigiosin, even making higher prodigiosin yield inside food lighter than the surface slick of prodigiosin. We mentioned it in the previous manuscript line 166-168. This is why some photos treated with excreta and a lower dose of AMP seemed more intense than those with WT larvae. However, we precisely quantified the prodigiosin yield inside food with the spectrophotometer, so we provided a prodigiosin yield following the photos of the slick. Therefore, we drew our conclusions mainly relying on the quantification of the prodigiosin yield. We actually used cecropin A for our experiments, so we added this information in the text. We hope that our replies can reignite your enthusiasm for our manuscript, and thanks for your great support!

**Reviewer #3 (Public Review):**
In this study, Wang and coworkers established a model of *Drosophila*-S. marcescens interactions and thoroughly examined host-microbe bidirectional interactions. They found that:(1) *Drosophila* larvae directly impact microbial aggregation and density;(2) *Drosophila* larvae affect microbial metabolism and cell wall morphology, as evidenced by reduced prodigiosin production and EPS production, respectively;(3) *Drosophila* larvae attenuate microbial virulence;(4) *Drosophila* larvae modulate the global transcription of microbes for adaptation to the host;(5) Microbial single-cell RNA sequencing (scRNA-seq) analysis revealed heterogeneity in microbial pathogenicity and growth;(6) AMPs are key factors controlling microbial virulence phenotypes.Taken together, they concluded that host immune factors such as AMPs are directly involved in the pathogen-to-commensal transition by altering microbial transcription.General comments:In general, this study is intriguing as it demonstrates that host immune effectors such as AMPs can serve as critical factors capable of modulating microbial transcription for host-microbe symbiosis. However, several important questions remain unanswered. One such question is: What is the mechanism by which AMPs modulate the pathogen-to-commensal transition? One hypothesis suggests that antimicrobial activity may influence microbial physiology, subsequently modulating transcription for the transition from pathogen to commensal. In this context, it is imperative to test various antibiotics with different modes of action (e.g., targeting the cell wall, transcription, or translation) at sub-lethal concentrations to determine whether sub-lethal doses of antimicrobial activity are sufficient to induce the pathogen-to-commensal transition.

Thank you for the important comments on our manuscript. We checked the effect of antibiotics (5 μg/μl kanamycin and 10 μg/μl ampicillin) on the virulence switch of *S. marcescens*. We found that the two antibiotics with the sub-lethal doses similarly resulted in a decrease in prodigiosin yield and virulence expression of *S. marcescens*. Intriguingly, the two antibiotics also resulted in a dramatic decline in the bacterial load and the expression of genes involved in cell growth. These results suggest that antibiotics reduced the virulence primarily through suppressing most activities of bacteria.

We found that larvae and AMPs at 40 μg/μl modestly resulted in a decrease in bacterial load and an increase in the relative level of genes involved in cellular proliferation, suggesting that AMPs could maintain the exponential phase of bacterial growth. This result is consistent that *Drosophila* larvae can support the long-term persistence of commensals in the shared habitat (DOI: 10.1016/j.cmet.2017.11.011). The inhibition could prevent bacteria from rapidly exhausting their nutritional resources, and consequently maintain symbiosis. It is likely that AMPs could maintain *S. marcescens* at the exponential phase of cell growth and prevent bacteria from rapidly exhausting their nutritional resources.

**Author response image 1. sa3fig1:** (A) Representative images of surface slick with *S. marcescens* alone, with kanamycin (5 μg/μl) and ampicillin (10 μg/μl). (B) The prodigiosin production of *S. marcescens* alone, with kanamycin (5 μg/μl) and ampicillin (10 μg/μl). n = 6 for each. (C) Bacterial loads of *S. marcescens* alone, with kanamycin (5 μg/μl) and ampicillin (10 μg/μl). n = 6 for each. (D, E) RT-qPCR analysis of the expression levels of downregulated and upregulated genes in the *S. marcescens* alone, with kanamycin (5 μg/μl) and ampicillin (10 μg/μl). n = 3 for each. Means ± SEMs. All variables have different letters, they are significantly different (*p* < 0.05). If two variables share a letter, they are not significantly different (*p* > 0.05). ns, no significance. Kruskal-Wallis test followed by Dunn’s multiple comparisons test.

**Recommendations for the authors:**

**Reviewer #1 (Recommendations For The Authors):**
Here are some specific points that need to be addressed:(1) Lack of statistical analysis for many figures. The authors should perform and report the statistical analysis for all figures where it is currently lacking, specifically, Figures 2C, D, E, F, H; Figures 3E, F; Figures 7G, H; Figure S2E, Figures S3D, E.

Thanks for your valuable suggestions. We re-checked the manuscript and performed the statistical analysis for these figures.

(2) For graphs showing dots, it should be specified what exactly individual dots show and how many animals were used per replicate. Also, time points at which specific analysis was performed should be specified.

We provided the important information in the legends in the revised manuscript.

(3) Figure 2. No letters illustrating statistical significance are shown, although this is claimed in the legend (line 848).

We added statistical significance in the updated Figure 2.

(4) In Figure 7, the authors used AMPs of defined concentration, but it is not specified what exactly these AMPs are. Please provide the full composition of the AMP mix used.

We used the antimicrobial peptide cecropin A produced by a silkworm. We added this information in the methods line 487-488 and Figure 7 legend.

(5) Figure S2B. To me, it looks like that medium with larvae is redder than after mechanical force. I find it hard to believe the quantification in panel C that the medium with larvae has 3 times less pigment as compared to the mechanical force.

Larvae could only agitate the surface of food (~0.4 cm), but sticks completely agitated the food up to 3 cm. Thus, the layer of food with pink pigment with agitation seemed much deeper than with larvae, which was responsible for the counterintuitively. We explained it in the previous manuscript (line 166-168). “Of note, the surface of the slick with agitation appeared lighter than that of larvae, mainly due to a stratification of prodigiosin following agitation.”

(6) The authors need to proofread the manuscript as there are missing words, terms that need definition, and wrong terms. For example, L86 - naked eye?, L117 - what do the authors mean by co-culture?, L309 - not resist but rather combat, L347 - Species? or competition?, Figure 2A - 2nd?

We have corrected these errors in the new manuscript. We added an "eye" in L86. Co-culture means “*S. marcescens* in co-culture”. Interspecies competition for nearly the same or similar nutrients and space occurs in the habitat.

(7) The authors should reorganize either the text or the figures' order in a way that the figures are described in a consecutive order (Figure 1A, B ... and not Figure 1D first and then 1A).

Thanks for your valuable advice. We reorganize the order of the text.

(8) Do the authors have an idea which bacteria they quantified in Figures 1E to 1G? I didn't find the medium that was used for culturing. Also, in Figure 1F, Is the control group comprised of females or males?

Mixed bacteria (bacteria in the living environment of *Drosophila*) were quantified in the NA medium that supports the growth of *Drosophila* microbiota (Jia Y, et al. Nat Commun. 2021) line 474-475. The control group comprised of both males and females with a 1:1 ratio. Similarly, the aged group contained 100 50-day-aged flies, male: female = 1:1. We provided details in Figure 1 legend line 849-850, 851-852.

(9) L118-129. it is not possible to make all these statements without any statistical analysis. To me, at 96h both treatments have the same CFUs, while the authors claim they are different.

We added statistical analysis in the current version. In fact, single *S. marcescens* became collapsed after 72 h post inoculation, and the CFU number of single *S. marcescens* declined step by step. The bacterial load of *S. marcescens* in co-culture was comparable (at 96 h post-inoculation, p>0.05) or higher (at 120 h post-inoculation, p＜0.001) than *S. marcescens* alone, possibly explained by the possibility that bacteria rapidly exhausted the nutritional resources and collapsed through population suicide. We rewrote this sentence line 125-129 in the updated manuscript.

(10) L136. term "symbionts" is not appropriate here.

We change “symbionts” into “*S. marcescens*”.

(11) In Figure 1, the authors used flies of different fitness: weak, strong, and infertile. They should be specific and describe exactly what these terms mean, are these mutants or treatments that affect the fitness?

We apologize for this missing information and add them in the method and legend. Strong flies (wild-type fly CS), weak flies (*yw; Sp/CyO; MKRS/TM6B*), infertile flies (*dfmr150M* null mutant) Figure 1 legend line 849-850.

(12) Figure S2. The title of this figure is misleading, please modify it. Mechanical force did affect S. marcescens but to a lesser degree as compared to larvae.

Thank you for your suggestion. We admit that mechanical force affected *S. marcescens* but to a lesser degree as compared to larvae, so we changed the title to "Biological factors mainly determine *S. marcescens* lifestyle."

**Reviewer #2 (Recommendations For The Authors):**
General improvement to writing and presentation (see below):Describing confluent growth would make more sense than 'slick' and then using descriptions of broken, etc. "colour intensity of the surface slick".

We used the slick to describe visible surface films of bacteria, which has been used in the previous study (DOI: 10.1038/s43705-023-00307-8). Slick is equal to confluent growth, but seems simple and easy than confluent growth. To make sense, we add this reference to the text.

We reorganized the text of Figure 1.

Suggest more specific language to describe observations. For example: Bacterial loading - S. marcescens growth (for example: the presence of dense fly populations reduced Sm growth).

Thanks for the suggests. We replaced some of them.

Symbiont, microbiota, microbiome, etc were all used interchangeably throughout the manuscript, but I am not sure I would call Sm part of the indigenous microbiome. Suggest to ensure proper usage and then harmonize throughout the ms.

We used microbes and microbiome to replace symbiont and microbiota, respectively.

Details missing from the message and Figure legends that would be helpful (including and especially Figure 7 - what AMP concentration?)

Thanks for valuable comments. According to this comment, we provided concrete details in the Materials and methods and Figure 7 legend about AMPs, including the source and concentration of AMPs line 487-488, 954-955. Please see the response below.

L73: define 'these issues" maybe or lead better with the prior sentence, it is not evident as currently written.

Change "to address these issues" to " To investigate whether and/or how the host modulates bacterial lifestyles,” and merge two paragraphs.

L74: repetitive sentence with the above.

Thanks for pointing out this detail. We deleted it.

L86: naked 'eye'.

Added.

L87: what is meant by 'weak flies'?

Genotypes were added in the updated manuscript. Weak fly stocks display weaker activity and generate fewer eggs than WT flies.

L96: bacterial load, not loading.

Corrected.

L128: no evidence to support, could be reflective of increased numbers in dying/dead larvae that impact total numbers in the vial.

The number of CFUs of *S. marcescens* alone was gradually decreased at 96 h post-inoculation. In addition, we observed pale biofilm on the surface of the medium at the late stage. The numbers of CFUs of *S. marcescens* alone at the later stages were reduced (compared to the peak load at 48 h post-inoculation), so it was deterred that bacteria could undergo ecological suicide. Ecological suicide of the bacterial population was similarly examined by recording the number of CFUs in the medium over time (Ratzke C, et al. Nat Ecol Evol. 2018.). Taken together, we draw a conclusion that bacteria possibly underwent ecological suicide.

L129: the prior sentence is in contradiction, reduced load only at early time points in the presence of larvae....

Thanks for pointing out this detail. We added " before 72 h post-inoculation " in the sentence.

L134: data is only focused on S marcescens, so inferring to 'symbionts' broadly is outside study.

We change “symbionts” into “*S. marcescens*”.

L139: sentence poorly written and confusing.

We re-organized this sentence.

To this end, we sought to examine the *S. marcescens* lifestyle switch from pathogenicity to commensalism by assessing the respective survival of flies on the fly medium that had been processed by single or coexisting *S. marcescens*.

L189: evidence for long-term symbiosis is not well established in this paper, suggest editing this language throughout to more specifically reflect what the data supports and leave such interpretations to discussion points and future work.

Thanks for your valuable advice. We deleted long-term and “thereby promoting the fitness of symbionts in the long maintenance.”.

L192; used metabolomics to assess the impacts of larvae on bacterial metabolism, as currently written does not make sense.

We rewrote this sentence. “Next, we investigated whether larvae could further elicit changes in the metabolism of *S. marcescens* using untargeted metabolomics.”

L331: the use of monitored here is not correct/odd.

We changed 'monitored' to 'reshaping’.

L340: While the authors initially see a cost to Sm in reduced load (CFUs) at 120 h populations associated with larvae become higher - there is also a cost to producing virulence factors, which their RNASeq and metabolomics data support - trade-offs between growth and virulence.

Thanks for your suggestion. We added “before 72 hours post inoculation” to define the early stage of the bacterial growth in the sentence.

**Reviewer #3 (Recommendations For The Authors):**
(1) Figures 1 A-D: What defines weak and strong flies, and what criteria determine the robustness of flies? How was the experiment conducted? The manuscript lacks details on this matter.

We thank you for your comments. We lack a criterium, but the robustness of flies comes from daily experience. Weak fly stocks display weak activity and generate fewer eggs than WT flies. Genotypes with different robustness were added in the legend in the updated manuscript

(2) The authors mentioned, "Noteworthily, the number of CFUs of S. marcescens alone was lower than S. marcescens in co-cultures at the late stage (at 96 h post inoculation), likely that bacteria rapidly exhausted their nutritional resources and underwent ecological suicide." How did they determine that the bacteria exhausted nutritional resources and underwent ecological suicide? One might speculate that larvae could have removed the bacteria simply by consuming them.

Thanks for this comment. Virtually, there were no larvae inside the vials with single *S. marcescens*, so bacterial cells were not consumed. However, the numbers of CFUs of *S. marcescens* alone at the later stages were reduced (compared to the peak load at 48 h post-inoculation), so it was deterred that bacteria could undergo ecological suicide. Ecological suicide of the bacterial population was examined by recording the number of CFUs in the medium over time (Ratzke C, et al. Nat Ecol Evol. 2018.). A similar method was also applied to the number of CFUs of *S. marcescens*. Taken together, we draw a conclusion that bacteria possibly underwent ecological suicide.

(3) Figure 2E: The experimental details should be provided in the text. What was the CFU of the bacteria used in this survival experiment?

We provided further experimental details in the legend line 869-870. The same amount of inocula was used in both single and coculturing *S. marcescens*.

(4) The experimental data in Figures 2G and 2H do not sufficiently prove the relationship between the width of the cell wall and virulence, as it lacks experimental validation.

Previous studies (DOI: 10.1371/journal.ppat.1005946) reveal that glucosylating toxins on the surface are primary virulence determinants, so an increased surface-anchored polysaccharide and protein profile promotes the virulence of the pathogen. Alterations in cell surface (the width of the cell wall) can be examined by TE. Moreover, TE was used to observe changes in the virulence of *S.* marcescens (DOI: 10.1093/nar/gkab1186). We think that the width of the cell wall could be used to reflect virulence in *S.* marcescens.

(5) While it's acknowledged that agitation decreases the color intensity of the bacteria, comparing mechanical agitation with larval crawling seems inappropriate, as the mechanical forces exerted by both methods are not of the same magnitude.

Thanks for the suggestion. In fact, food was agitated more heavily by glass sticks than by larvae, because larvae merely agitated the surface of food (about 0.5 cm-depth). If the decrease in bacterial load and color was related to the magnitude of agitation, larvae would confer a less decrease (from the decrease in stick agitation) in bacterial load than the sticks. Consequently, it would further support our result that biofactors more importantly confer the inhibition of *S. marcescens* than force.

(6) Figure 4D: with this metabolome data, they mentioned, "host suppresses differentiation of S. marcescens into the population with pathogenicity." What evidence supports the claim that downregulation of amino acid metabolism, phosphotransferase system, and ABC transporter directly correlates with decreased pathogenicity?

Thanks for the comment. Earlier studies showed that amino acid-derived quorum sensing molecules are closely related to bacterial pathogenicity (Defoirdt T. PLoS Pathog. 2019; Wen J, et al. Microbiol Spectr. 2022). Moreover, the phosphotransferase system and ABC transporter can transport and/or produce virulence factors. Therefore, we claimed that downregulation of amino acid metabolism, phosphotransferase system, and ABC transporter directly were related to decreased pathogenicity. To support this claim, we add some references in the updated manuscript line 662-664, 827-830.

(7) Serotonin: Does serotonin also reduce the virulence of S. marcescens?

Our primary result showed that serotonin indeed could reduce the virulence of *S. marcescens* (figure supplement 4), because the survival rate of adult flies was increased and the expression levels of virulence-related genes of *S. marcescens* alone in the case of serotonin.

(8) Figures 6D, E, H, I: The expression of key genes should be verified using quantitative real-time polymerase chain reaction (qRT-PCR), as scRNA-seq expression levels might not accurately reflect the true expression levels.

Bacterial single-cell RNA-seq can evaluate alterations in gene expression in the single-cell resolution. The expression of key genes screened by scRNA-seq was changed only in subpopulations, so the average expression of these genes would be comparable when mixed with a large population. We are afraid that qRT-PCR could be illegible to verify the expression of genes in subpopulations.

(9) Figure 7: The authors mentioned. "AMPs were supplemented to fly food". However, I could not find information regarding which AMPs and their respective concentrations (i.e., concentration of each AMP) were used in this study. This is a critical aspect of the research; therefore, details should be provided.

Thanks for your important suggestions. We used the antimicrobial peptide cecropin A, which is produced by silkworms. We provided this information in the methods line 487-488. The concentrations of cecropin A were added in Figure 7 legend.

(10) Figure 7: Delta AMP + AMP exhibited a stronger effect on the bacteria compared to AMP alone, indicating that immune effectors other than AMP may be involved. Since the IMD pathway is necessary for most immune effectors, including AMP, it would be interesting to test IMD pathway mutant animals and compare them with Delta AMP. Delta AMP + AMP exhibited a stronger effect on the bacteria compared to AMP alone.

We appreciate this important question. Indeed, Delta AMP + AMP exhibited a stronger effect on the bacteria compared to AMP alone. We admitted that immune effectors other than AMP may be involved. Alternatively, mechanical force, to a less extent, accounted for the stronger effect on the bacteria (Explained by larvae agitation in figure supplement 2). To rule out this possibility, we examined the effect of total immune effectors on the bacterial load and the prodigiosin yield of *S. marcescens* using the IMD pathway mutant (*RelE20* larvae). Our result showed that the optical density and yield of prodigiosin in Delta AMP group did not significantly differ from the ones in *RelE20* group. Moreover, the load of *S. marcescens* associated with Delta AMP mutant was comparable to that of *S. marcescens* associated with *RelE20* mutant. These results suggested that AMPs play a major role in recapitulating the response of _S. marcescens t_o larvae.

“To rule out the potential role of other immune effectors, we turned to the IMD pathway mutant *RelE20* that is deficient in total immune effectors. Our result showed that the optical density and yield of prodigiosin in *RelE20* group did not significantly differ from the ones in *DAMP* group (figure supplement 7A, B). Moreover, the load of *S. marcescens* associated with *RelE20* mutant was comparable to that of *S. marcescens* associated with Delta AMP mutant (figure supplement 7C).”

We now added these results in the text line 326-331.